# THIN-THICK ADAPTER: SEGMENTING THIN SCANS USING THICK ANNOTATIONS

## ABSTRACT

Medical imaging segmentation has emerged as a critical field within medical analysis, predominantly operating on thicker CT slices ($> 1$mm, usually 5mm) derived from post-processed versions of original thinner slices (generally $\leq 1$mm). While thin slices offer superior spatial resolution and diagnostic precision for clinicians, the scarcity of annotated data for supervision limits segmentation to thicker slices. Acquiring thin slice annotations for training a dedicated segmenter is resource and time-intensive, often requiring an order of magnitude more effort compared to annotating thick slices, making it impractical to accumulate sufficient high-quality thin annotations for robust supervised models. Furthermore, directly employing thick data to train models for thin slices faces significant domain gaps in scale. In response to these multifaceted challenges, we introduce three key contributions. Firstly, we propose a novel task of **segmenting thin scans using annotations from thicker slices**, addressing a practical clinical need. Secondly, we present the **CQ500-Thin** dataset, comprising Non-Contrast CT scans featuring Intracranial Hemorrhage (ICH) and expertly-labelled pixel-level thin annotations tailored for evaluation, accompanied by corresponding benchmarks and evaluation metrics. Lastly, we introduce the **Thin-Thick Adapter (TTA)**, a simple yet highly effective module that plays a pivotal role in bridging the domain gap between thin and thick scans. TTA has notably enhanced segmentation performance, boosting mDSC by $10.18\%$ and mIoU by $12.34\%$ compared to the vanilla nnUNet baseline. This substantial improvement extends the applicability of segmentation models, enabling their use in various clinical scenarios, including unsupervised approaches, and thus advancing the cutting-edge of medical imaging segmentation research. The code and dataset will be published under acceptance.

## 1 INTRODUCTION

Segmentation has been a key focus of computer vision for an extended period and has important applications in medical imaging (Long et al., 2015). Medical images, derived from predominant modalities such as Computed Tomography (CT), Positron Emission Tomography (PET), and Magnetic Resonance Imaging (MRI), are inherently volumetric in nature, encompassing a three-dimensional (3D) spatial representation (Singh et al., 2020). Consequently, the precision and accuracy of 3D segmentation is indispensable for tasks ranging from disease diagnosis to treatment planning.

Furthermore, among diagnostic imaging modalities, CT is one of the most commonly employed techniques (Muir & Santosh, 2005). CT scans produce two discernible image types based on slice thickness: thick slices and thin slices (Huang et al., 2021). Thick slices are derived either from thick scan acquisitions or by retrospective reconstruction of thin slices (Boxwala & Rosenman, 1994), while thin slices are acquired directly through thin scanning procedures. During the CT scanning process, the patient is positioned on the CT scanner table and gradually moved through the gantry, which houses the X-ray tube and detector array. This rotation and movement of the table are controlled by the pitch parameter (Wang & Vannier, 1999), which determines the distance the CT scanner table travels during each rotation of the X-ray tube. If the pitch number is greater than 1, it means that the table travels more than the width of the beam, i.e. there are gaps. In contrast, if the pitch number is less than 1, it means that the table travels less than the width of the beam, i.e. there is overlap. During prospective reconstruction, the pitch plays a crucial role

in determining the thickness of the resulting CT slices. A smaller pitch value indicates a smaller distance traveled by the table per rotation, resulting in thinner CT slices. This parameter directly influences the spatial resolution and image quality of the CT scans. The thin slices, in another words, is obtained from the prospective reconstruction of thin scan. After acquiring thin slices, a retrospective reconstruction (Boxwala & Rosenman, 1994) technique can be employed to obtain thick slices. This process involves utilizing an averaging algorithm, such as the average intensity projection (AIP) (Shirai et al., 2012), to merge multiple thin slices into a single thick slice. The thick slices can be generated with varying thicknesses and from different orientations, including axial, sagittal, and coronal planes (van Waes et al., 1983). Additionally, there are three different interval strategies for averaging thin slices into thick slices: contiguous, non-contiguous, and overlapped interval.

Thick slices exhibit diminished resolution along the depth axis, commonly referred to as z-resolution. In contrast, thin slices encapsulate more detailed volumetric information, thus offering superior capabilities for diagnosis. Unfortunately, publicly accessible CT datasets, accompanied by pixel-wise ground truth annotations, predominantly provide thick-slice acquisitions. However, there is a domain gap between thin and thick scans. The averaging nature of thick scans (compared to thin) can alter the noise statistics of images, as well as blur anatomic structures. Consequently, existing medical imaging segmentation methods underperform in thin-slice segmentation. As afore-mentioned, the manual annotation CT scans by an expert human annotator, such as a radiologist, is onerous given the laborious and time-intensive nature of this process. This challenge is further exacerbated when it comes to annotating thin slices, which involve a larger number of scans.

We chose the brain hemorrhage dataset for our study because it provided convenient access to both multi-semantic thick scans and unlabeled thin scans, some of which we already possessed.

Thus, the development of framework for segmenting thin slices based on thick-slice annotations is required to maximize the utility of thin slice acquisitions. In light of this, this paper introduces the following three key contributions as follows.

- We present an innovative problem formulation and scenario: the segmentation of exclusively thin slices, utilizing annotations originally designed for thick slices, without necessitating any paired annotations encompassing both thin and thick slices.
- We have released a dataset called **CQ500-Thin**, derived from the publicly available CQ500 dataset (Chilamkurthy et al., 2018). It comprises 374 thin Non-Contrast CT scans along with 15 expertly labelled pixel-level annotations for evaluation purposes. This dataset can also be utilized as a benchmark for future research.
- We have introduced a robust pipeline referred to as the **Thin-Thick Adapter**. This pipeline employs a straightforward yet effective data alignment technique, acting on thick slices to harmonize their shapes with thin slices. Additionally, we employ the 3D Cross Pseudo Supervision (3D-CPS) (Chen et al., 2021b; Huang et al., 2022) on nnUNet (Isensee et al., 2021) for unsupervised domain adaptation (UDA) to enhance the model's performance on unlabeled thin slices, achieving superior results compared to existing methods in the context of thin slice segmentation tasks. Moreover, this pipeline serves as a sturdy baseline for addressing the problem at hand and lays the groundwork for future research endeavors in this domain.

## 2 RELATED WORKS

**3D Medical Imaging Segmentation** Transformer architectures, such as TransUNet (Chen et al., 2021a), Medical Transformer (Valanarasu et al., 2021), and TransBTS (Wang et al., 2021), are emerging as strong contenders to CNNs in medical image segmentation. They leverage self-attention to capture global contexts in medical images, transcending CNN's local limits. Hybrid models like TransFuse (Zhang et al., 2021) and CoTr (Xie et al., 2021) blend Transformer and CNN strengths. Here, CNNs extract local features, while Transformers address global contexts, enriched by skip connections. Strategies such as proxy task pre-training in Swin UNETR (Hatamizadeh et al., 2021) optimize Transformer performance on smaller datasets.

On the other hand, CNN-based designs, like nnUNet (Isensee et al., 2021) and UNETR (Hatamizadeh et al., 2022), remain dominant due to their U-Net adaptations. For instance, UC-

TransNet (Wang et al., 2022) modifies skip connections with attention. nnFormer (Zhou et al., 2021) smoothly incorporates convolutional layers within its Transformer design, leveraging encoders like Swin Transformers in Swin UNETR (Hatamizadeh et al., 2021). This facilitates capturing hierarchical and volumetric representations, with self-supervised pre-training benefits echoing those in Transformer models. Historically, research primarily applied 2D methods to reconstruct 3D volumes slice-by-slice, leading to inconsistencies. Most methods focus on thick slice segmentation, limiting their adaptability to thin slices. Our Thin-Thick Adapter addresses this by using thick annotations for thin-slice segmentation.

**Interpolation and Super-resolution** In medical imaging, especially MRI, interpolation and super-resolution techniques have advanced significantly. The study by (Thanh & Hai, 2017) presents a 3D image construction from 2D MRI cortex images using a trilinear interpolation after preprocessing and applying the multilevel Otsu method for brain extraction. Conversely, (Ghoshal et al., 2023) introduces a technique using edge-preserved kriging interpolation for missing slice prediction, speeding up the process through parallel processing and then applying shearlet transform and the marching cubes method for visualization. This approach considerably accelerates reconstruction, even for large datasets.

Regarding super-resolution, (Brudfors et al., 2018) presents the UniRes generative model, which treats high-resolution image recovery as an inverse problem. Incorporating a multi-channel total variation prior, it exhibits enhanced MRI super-resolution capabilities. (Pinaya et al., 2022) harnesses Latent Diffusion Models for synthetic high-resolution 3D brain image generation, effectively using the UK Biobank dataset for realistic data generation with variable conditioning. Finally, (Lyu et al., 2020) proposes a method that integrates multiple contrast images in a high-level feature space for superior super-resolution, especially effective when starting with significantly down-sampled images.

**Unsupervised Domain Adaptation** In the arena of unsupervised domain adaptation (UDA) and deep learning robustness, several notable advancements have been made. The SDC-UDA (Shin et al., 2023) study introduces an innovative framework for slice-direction continuous cross-modality medical image segmentation. SDC-UDA leverages intra- and inter-slice self-attentive image translation, coupled with uncertainty-constrained pseudo-label refinement and volumetric self-training. Unlike previous UDA techniques in medical imaging, SDC-UDA ensures continuous segmentation in the slice direction, enhancing its accuracy and clinical utility. This method demonstrated superior slice-direction continuity and state-of-the-art segmentation performance across multiple datasets. In a separate vein, CANARY (Sun et al., 2023) addresses the challenge of evaluating the adversarial robustness of deep learning models. The CANARY platform utilizes a common scoring system, incorporating four dimensions and 26 (sub)metrics. It employs a two-way evaluation strategy and introduces Item Response Theory (IRT) to ensure fairness in scoring, thereby providing comprehensive evaluations of models or attack/defense algorithms. Lastly, the study (Yao et al., 2023) presents an augmentation technique tailored for the Segment Anything Model (SAM) to enhance its performance in segmenting noisy, low-contrast medical images. This method integrates multi-box prompt augmentation with an aleatoric uncertainty-based false-negative and false-positive correction strategy. Further, the introduction of the Single-Slice-to-Volume (SS2V) method allows 3D pixel-level segmentation from just a single 2D slice annotation, emphasizing SAM's adaptability even in challenging medical imaging scenarios.

## 3 Unpaired Thin-Slice Segmentation Using Thick Annotations

As previously mentioned, thin slices and thick slices belong to distinct domains, with thin slices often outperforming thick slices in various aspects. Consequently, there is a need to focus on segmenting thin slices exclusively. To address this, we have introduced a novel problem formulation: segmenting thin slices using only thick annotations, without the requirement for paired thin and thick slices or annotations. The necessity for this proposed problem formulation is outlined below.

**Utilizing Thick Annotations** We opted to utilize thick annotations instead of thin annotations due to the prohibitively high cost associated with pixel-level annotations for 3D CT volumes. Annotating thin slices at the pixel level is considerably more expensive than annotating thick slices. Con-

sequently, all existing publicly available pixel-level annotated Non-Contrast CT datasets primarily consist of thick slices, until the introduction of our CQ500-Thin dataset. Because the quantity of thin annotations is quite limited, both fully supervised training with thin annotations and pre-training on thick annotations followed by fine-tuning on thin annotations cannot attain optimal performance, which is demonstrated in Table 3. Furthermore, current 2D and 3D baseline models, trained on thick annotations, exhibit suboptimal performance. For instance, 2D segmentation methods like SegViT (Zhang et al., 2022; 2023) suffer from spiny and inconsistent predictions when viewed from sagittal and coronal perspectives. This inconsistency arises because they fail to learn the coherence and inter-slice relationships, resulting in masks that appear jagged. On the other hand, 3D segmentation methods like nnUNet (Isensee et al., 2021) encounter issues with blocky predictions when observed from sagittal and coronal orientations. This occurs due to their training on thick slices, which possess a different thickness compared to the thin slices. Thick slices and annotations are better suited for coarse-grained data, whereas thin slices are better for fine-grained data.

**Utilizing Unpaired Data**  We employ unpaired data because the availability of datasets containing multi-semantic pixel-level annotations is notably scarce. Most of these datasets exclusively annotate thick slices. In fact, datasets with pixel-level annotations on thin slices are nearly nonexistent, let alone those containing paired thin-thick data. Consequently, addressing the challenge of thin-slice segmentation cannot be accomplished through joint-loss or registration methods, as there is a lack of paired thin-thick datasets. Consequently, utilizing unpaired datasets has enhanced the versatility, universality, and generalizability of our approach.

**Utilizing Domain Adaptation Approach**  The key difference between thin and thick slices isn't the number of slices, but rather the thickness of each individual slice, which collectively affects the depth resolution. Attempting to use thick slices to segment thin ones presents challenges due to issues such as partial volume artifacts, reduced spatial resolution, and noise averaging. This essentially means that multiple structures can be contained within a single voxel, leading to a loss of contrast resolution for small objects (partial voluming) and blurred edges for larger objects. Moreover, the images display differing noise statistics, resulting in a domain shift. In essence, the problems of losing contrast resolution for small objects and blurring the edges of larger objects are interconnected.

## 4 DATASET

**CQ500-Thin**  CQ500-Thin is a dataset dedicated to Intracranial Hemorrhage, a subset derived from the publicly available CQ500 dataset (Chilamkurthy et al., 2018). It consists of 374 thin non-contrast CT scans, and we added pixel-level semantic segmentation annotations to 15 randomly selected scans. These annotations cover five semantic categories: Epidural Hemorrhage (EDH), Intraparenchymal Hemorrhage (IPH), Intraventricular Hemorrhage (IVH), Subarachnoid Hemorrhage (SAH), and Subdural Hemorrhage (SDH). The annotation process involved contributions from a neurologist, a junior doctor, and a medical student, with review by a radiologist.

**ROTEM-Thin**  ROTEM-Thin originates from a private dataset within the ROTEM project, overseen by three doctors, comprising 389 thin non-contrast CT scans. We added pixel-level semantic segmentation annotations by randomly selecting 10 images, including the same five semantics as the CQ500-Thin dataset. This dataset serves as an internal benchmark to enhance the credibility and robustness of our proposed method.

## 5 METHODOLOGY

### 5.1 DATA ALIGNMENT

Let's assume that for a single thick scan, its shape is denoted as $(c, h, w, d)$. Please note that, in the case of a Non-Contrast CT protocol, it is typical to have a single channel ($c = 1$). Additionally, for a head CT scan, the dimensions are generally set to $h = w = 512$ (Goldman, 2007; Wu et al., 2020).

The spacing, with respect to height, width, and depth, is defined as $(s_h, s_w, s_d)$. Note that, in the case of a Non-Contrast CT scan, usually the $s_h = s_w \leq 1$mm (Goldman, 2007; Lolli et al., 2016).

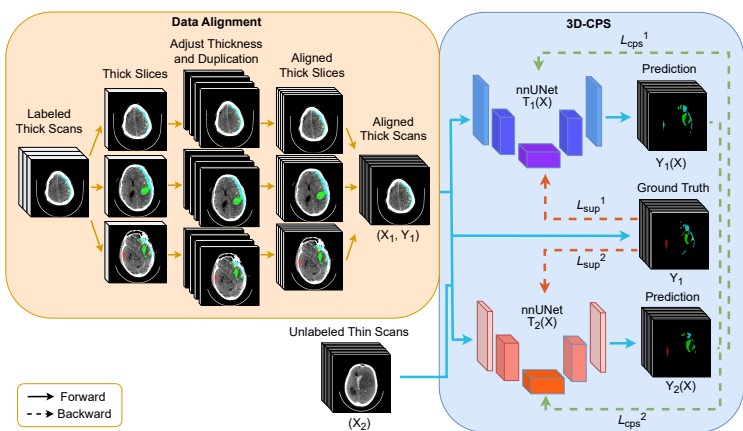

Figure 1: This diagram illustrates the workflow of the Thin-Thick Adapter. Initially, labeled thick scans undergo thickness adjustment and duplication $n$ times, resulting in aligned thick scans. These scans are subsequently used to train nnUNet as a supervision. The addition of 3D-CPS and unlabeled thin data further enhances the performance of the Thin-Thick Adapter.

Hence, we can set $s_{thin} = s_h = s_w$. And the spacing in depth $s_d$ is the thickness of slices. Therefore, we define the alignment coefficient as

$$n = \lceil \frac{s_d}{s_{thin}} \rceil \tag{1}$$

We can now duplicate each slice in the thick scans $n$ times and adjust the depth spacing to $s_{thin}$. As a result, the aligned thick data will have the shape $(c, h, w, nd)$, and the spacing will be $(s_{thin}, s_{thin}, s_{thin})$.

## 5.2  3D-CPS

The CPS is originally designed for 2D semi-supervised semantic segmentation (Chen et al., 2021b), it is adaptable for 3D unsupervised domain adaptation (Huang et al., 2022). The objective of a unsupervised domain adaptation (UDA) task is to train a segmentation network by utilizing both labeled $D_l$ and unlabeled $D_u$ data. The proposed method comprises two parallel segmentation networks, denoted as $FT_1$ and $T_2$. These networks share the same architecture but have different initial weights, represented as $\theta_1$ and $\theta_2$.

The input data $X$ for UDA training undergoes preprocessing using the same pipeline as nnU-Net (Isensee et al., 2021), followed by applying the default data augmentation from nnU-Net. $T_1(X)$ and $T_2(X)$ generate predicted one-hot confidence maps based on the two parallel networks, while $Y_1(X)$ and $Y_2(X)$ produce predicted one-hot label maps derived from $T_1(X)$ and $T_2(X)$, respectively. The training objective consists of two key components: the supervision loss $L_{sup}$ and the cross-pseudo supervision loss $L_{cps}$, both are the combination of dice and cross-entropy losses, which are aligned with the default configured of nnU-Net's joint loss. The cross-pseudo supervision loss, $L_{cps}$, comprises two parts: $L_{cps}^l$ and $L_{cps}^u$, incorporating the CPS loss on labeled and unlabeled datasets. In this setup, the two parallel networks consider the output of the other network as their own pseudo-labels. The losses are formulated as:

$$L_{sup} = l_{sup}(T_1(X), Y) + l_{sup}(T_2(X), Y) \tag{2}$$

$$L_{cps} = L_{cps}^l + L_{cps}^u \tag{3}$$

$$L_{cps}^l = l_{cps}(T_1(X), Y_2) + l_{cps}(T_2(X), Y_1), X \in \text{labeled data} \tag{4}$$

$$L_{cps}^u = l_{cps}(T_1(X), Y_2) + l_{cps}(T_2(X), Y_1), X \in \text{unlabeled data} \tag{5}$$

$$L = L_{sup} + \lambda L_{cps} \tag{6}$$

where $\lambda$ serves as a hyper-parameter that must be predefined to determine the weight of the cross-supervision loss. We set $\lambda$ to increase linearly from 0 to 0.5, and it remains fixed after a specific epoch $\epsilon$, as opposed to having a constant value as in CPS.

## 6 EXPERIMENTS

### 6.1 EXPERIMENT SETUP

The Thin-Thick Adapter experiments consist of several steps. First, data alignment is applied to 191 pixel-level annotated thick slices from the BHSD dataset (Wu et al., 2023). Subsequently, this aligned thick data is combined with unlabeled data from various sources, including CQ500-Thin (359 unlabeled thin scans), ROTEM-Thin (379 unlabeled thin scans), or a combination of both (738 unlabeled thin scans). These datasets are utilized to perform 3D-CPS (Chen et al., 2021b; Huang et al., 2022) on the nnUNet backbone (Isensee et al., 2021), resulting in three distinct sets. The evaluation is conducted on sets of 15, 10, and 25 pixel-level annotated scans, respectively. We set $\lambda$ to increase linearly from 0 to 0.5, and it remains fixed after 500 epochs.

To ensure a fair comparison, the baseline approach involves training the nnUNet model on the 191 thick annotated scans from the BHSD dataset. Subsequently, this trained model is directly applied for inference on 15, 10, and 25 pixel-level annotated scans from CQ500-Thin, ROTEM-Thin, and both datasets, respectively, without any adaptation.

An alternative approach that may tackle the challenge of thin slice segmentation, as mentioned earlier, is to employ a state-of-the-art 2D semantic segmentation method. This involves training and inferring CT scans on a slice-by-slice basis and subsequently reconstructing the individual slices to form a complete 3D CT volume. In a manner similar to the 3D baseline (nnUNet), we disassembled 191 pixel-level annotated thick scans into individual slices and trained SegViT (Zhang et al., 2022) on these thick slices. Subsequently, we evaluated the performance of the trained model on thin evaluation sets.

We also applied two techniques, trilinear interpolation (Thanh & Hai, 2017) and UniRes (Brudfors et al., 2018), for super-resolution. Initially, these techniques were used to enhance the resolution of 191 thick scans from the BHSD dataset, making them isotropic among all three dimensions. In other words, we adjusted the spacing in the depth (slice thickness) to be consistent with their height and width. We also employed these techniques on the corresponding annotations, ensuring that the semantic content of the annotations remained unchanged. To achieve this, we used the semantics from the original annotations as a reference during the calibration process. Subsequently, we used these super-resolution 191 thick scans to train the nnUNet and assessed their performance on the three evaluation sets. The difference between the thick scans, trilinear interpolation, UniRes, aligned thick scans, and the corresponding thin scans, has been shown in Figure 2. It is obvious that interpolation and super-resolution techniques such as trilinear interpolation and UniRes introduce blur compared with original scans, and thick scans has average out noise than thin scans due to average intensity projection (AIP) (Shirai et al., 2012), as we mentioned in Section 1 and 3.

We selected nnUNet (Isensee et al., 2021) as our baseline because it is a convolutional architecture that surpasses transformer-based or convolutional-transformer hybrid architectures when working with limited data (Wu et al., 2023) due to its inherent CNN attributes, including locality and translation equivariance (Dosovitskiy et al., 2020). Furthermore, we chose 3D-CPS (Huang et al., 2022) as our unsupervised domain adaptation (UDA) technique because it outperforms other semi-supervised techniques (Wu et al., 2023) such as Entropy Minimization (Grandvalet & Bengio, 2004), Mean Teacher (Tarvainen & Valpola, 2017), or Interpolation Consistency (Verma et al., 2022).

All experiments were conducted on a hardware platform featuring two Intel Xeon Platinum 8360Y 2.40GHz CPUs, 8 NVIDIA A100 40G GPUs, and 256GB of RAM.

### 6.2 EVALUATION METRICS

**mDSC and aDSC**  The mean Dice Similarity Coefficient (mDSC) is the mean of Dice Similarity Coefficients (DSCs) among all semantics, which is a commonly utilized evaluation metric in medical imaging segmentation applications. It quantifies the degree of overlap between the predicted segmentation $P$ and the corresponding ground truth segmentation $G$. The mDSC value ranges between 0 and 1, with 0 indicating no overlap and 1 indicating a perfect match between the predicted and ground truth segmentations. Assume we have $n$ semantics, mDSC $= (\sum_{i=1}^{n} \frac{2PG}{P+G})/n$

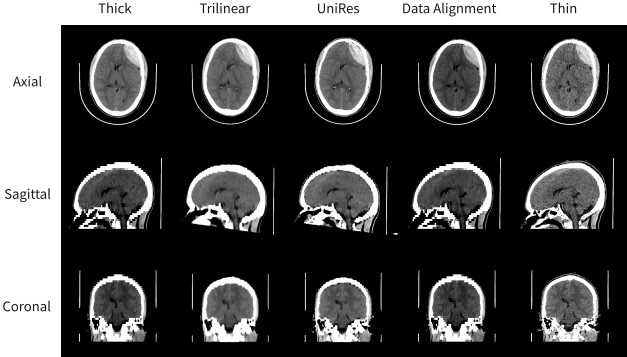

Figure 2: The comparison between orginal thick scan, super-resolution techniques, data alignment techinque in our TTA, and the original thin scan. Please note that we did not use any paired thick and thin scans, this is just for demonstration.

In contrast, the class-agnostic Dice Similarity Coefficient (aDSC) is consider all the semantics as a single foreground semantic, essentially measuring the overall quality of segmentation regardless of specific foreground categories.

The aDSC is particularly valuable in assessing the accuracy of segmentation models, especially in situations where there is a substantial class imbalance. It provides a comprehensive measure of the segmentation performance, accounting for both the presence and absence of the segmented regions.

**mIoU and aIoU**   The mean Intersection over Union (mIoU) is similar to the mDSC, which involves determining the intersection of the predicted and ground truth segmentations and dividing it by the union of their respective areas. The intersection represents the common region between the two segmentations, while the union represents the combined area of both the predicted and ground truth segmentations. It measures the degree of overlap between predicted $P$ and ground truth $G$ segmentation masks, providing an overall assessment of the model's performance across all semantics. Assume we have $n$ semantics, mIoU $= (\sum_{i=1}^{n} \frac{P \cap G}{P \cup G})/n$

The class-agnostic Intersection over Union (aIoU) treats all semantics as a unified foreground semantic. This essentially quantifies the overall segmentation quality without distinguishing specific foreground categories.

## 6.3   RESULTS

The results demonstrate that our proposed Thin-Thick Adapter (TTA) significantly outperforms all other methods, including the 3D nnUNet baseline, 2D SegViT, trilinear interpolation, and UniRes super-resolution, as shown in Table 1. Additionally, TTA effectively resolves the issues of blocky and spiny masks observed in the sagittal and coronal views, which are present in nnUNet and SegViT, and produces accurate predictions compared to trilinear interpolation and UniRes, as illustrated in Figure 3.

| | CQ500-Thin | | | | ROTEM-Thin | | | | Both | | | |
| --- | --- | --- | --- | --- | --- | --- | --- | --- | --- | --- | --- | --- |
| | mDSC | aDSC | mIoU | aIoU | mDSC | aDSC | mIoU | aIoU | mDSC | aDSC | mIoU | aIoU |
| nnUNet | 70.28 | 77.80 | 55.03 | 63.66 | 47.30 | 67.98 | 34.18 | 51.50 | 66.76 | 75.11 | 51.11 | 60.14 |
| SegViT | 69.62 | 76.79 | 54.26 | 62.32 | 38.71 | 65.40 | 26.32 | 48.59 | 63.73 | 73.40 | 47.79 | 57.98 |
| Trilinear | 64.99 | 74.13 | 48.74 | 58.89 | 42.40 | 64.89 | 29.61 | 48.03 | 62.64 | 71.55 | 46.17 | 55.70 |
| UniRes | 67.43 | 73.97 | 51.52 | 58.70 | 42.04 | 62.92 | 28.92 | 45.90 | 63.53 | 70.84 | 47.19 | 54.85 |
| **TTA (Ours)** | **77.54** | **83.62** | **64.33** | **71.86** | **59.69** | **75.64** | **47.68** | **60.82** | **76.94** | **81.36** | **63.45** | **68.57** |

Table 1: The comparative results clearly indicate that the proposed Thin-Thick Adapter outperforms the 3D baseline, 2D segmentation method, as well as interpolation and super-resolution techniques by a significant margin.

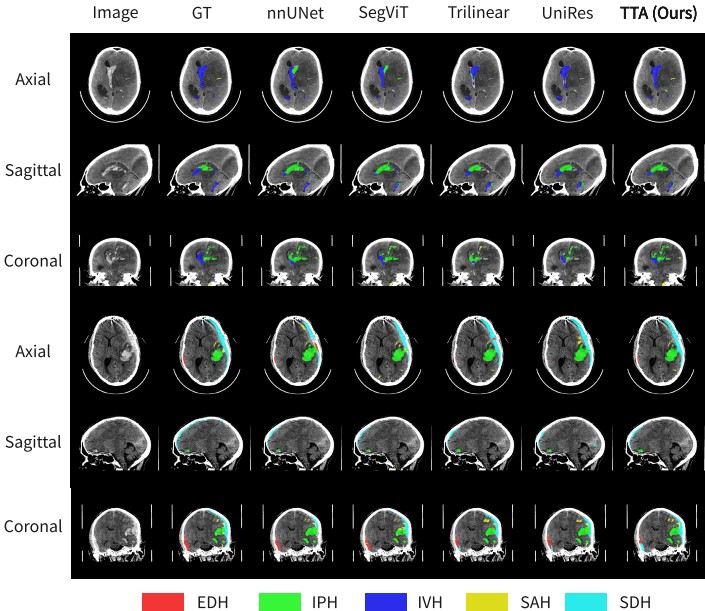

Figure 3: The visual comparison presents inference masks generated by various methods alongside the ground truth (GT). It is evident that the nnUNet baseline's inference mask in the sagittal and coronal planes appears blocky, while the SegViT 2D baseline's inference mask in these planes seems spiny. Notably, our TTA consistently outperforms all methods across axial, coronal, and sagittal views.

## 6.4 ABLATION STUDIES

In this section, we conduct a thorough assessment of our Thin-Thick Adapter (TTA) across various settings, shown in Table 2. These ablation studies are designed to elucidate the impact of each component within the pipeline on the overall performance.

In the first experiment, we trained the nnUNet baseline on the 191 thick scans from BHSD and assessed its performance on 15, 10, and 25 thin scans from CQ500-Thin, ROTEM-Thin, and both, respectively. In the second experiment, we exclusively applied data augmentation (DA) to the 191 thick scans without any other techniques, and then evaluated the model on the same three evaluation sets. In the third experiment, we employed 3D-CPS with both the original 191 thick scans and the unlabeled scans from CQ500-Thin (359 thin scans), ROTEM-Thin (379 thin scans), and both (738 unlabeled thin scans) to train the nnUNet backbone, without implementing additional techniques. We evaluated this model on the previously mentioned evaluation sets. Finally, we used the proposed Thin-Thick Adapter, which trained on the aligned 191 thick scans with 359, 379, and 738 unlabeled thin scans respectively, and then assessed its performance on the thin evaluation sets.

The results demonstrate that the data alignment technique significantly contributed to our proposed method and cannot be substituted by unsupervised domain adaptation (UDA) techniques such as 3D-CPS. Furthermore, this ablation study shows that both data alignment and 3D-CPS techniques play a role in the effectiveness of the proposed Thin-Thick Adapter.

| | CQ500-Thin | | | | ROTEM-Thin | | | | Both | | | |
|---|---|---|---|---|---|---|---|---|---|---|---|---|
| | mDSC | aDSC | mIoU | aIoU | mDSC | aDSC | mIoU | aIoU | mDSC | aDSC | mIoU | aIoU |
| nnUNet | 70.28 | 77.80 | 55.03 | 63.66 | 47.30 | 67.98 | 34.18 | 51.50 | 66.76 | 75.11 | 51.11 | 60.14 |
| + DA | 76.48 +6.2 | 81.14 +3.3 | 62.95 +7.9 | 68.26 +4.6 | 58.34 +11.0 | **77.03** +9.1 | 46.23 +12.1 | **62.64** +11.1 | 74.73 +8.0 | 79.95 +4.8 | 60.62 +9.5 | 66.60 +6.5 |
| + 3D-CPS | 70.89 +0.6 | 78.67 +0.9 | 55.91 +0.9 | 64.67 +1.0 | 48.47 +1.2 | 69.01 +1.0 | 35.37 +1.2 | 52.91 +1.4 | 67.40 +0.6 | 76.09 +1.0 | 52.03 +0.9 | 61.37 +1.2 |
| **TTA (Ours)** | **77.54** +7.3 | **83.62** +5.8 | **64.33** +9.3 | **71.86** +8.2 | **59.69** +12.4 | 75.64 +7.7 | **47.68** +13.5 | 60.82 +9.3 | **76.94** +10.2 | **81.36** +6.3 | **63.45** +12.3 | **68.57** +8.4 |

Table 2: The results emphasize the vital role of data alignment in our proposed method, which cannot be replaced by unsupervised domain adaptation (UDA) techniques like 3D-CPS. Additionally, the ablation study highlights the joint contributions of data alignment and 3D-CPS techniques to the effectiveness of the Thin-Thick Adapter.

In the second ablation study presented in Table 3, our aim is to investigate the performance of two approaches in thin-slice segmentation when compared to our Thin-Thick Adapter. For this purpose, we randomly selected 5 volumes from the 25 annotated thin slices as the evaluation set.

In the first experiment, we initially trained nnUNet on 191 thick scans from BHSD and then assessed its performance on the 5 selected thin scans. In the second experiment, nnUNet was directly trained on the remaining 20 thin scans and evaluated on the same 5 thin scans. In the third experiment, nnUNet was first pre-trained on the 191 thick scans from BHSD, followed by fine-tuning on the 20 thin scans. The evaluation was conducted on the 5 thin scans in the evaluation set. In the last experiment, our proposed Thin-Thick Adapter was trained using the 191 labeled thick slices from BHSD and 738 unlabeled thin scans from both CQ500-Thin and ROTEM-Thin. Subsequently, we evaluated its performance on the same 5 thin scans.

Notably, the results clearly demonstrate the superior performance of our Thin-Thick Adapter compared to other methods. This ablation study reaffirms our earlier claim in Section 3 that, due to the limited number of annotated thin scans, achieving satisfactory performance is challenging with either supervised methods on thin scans or fine-tuning on them.

|  | mDSC | aDSC | mIoU | aIoU |
|---|---|---|---|---|
| nnUNet (Thick) | 60.86 | 74.44 | 45.27 | 59.29 |
| nnUNet (Thin) | 50.24 | 79.18 | 38.70 | 65.54 |
| nnUNet (Fine-tune) | 51.28 | 73.38 | 38.03 | 57.95 |
| **TTA (Ours)** | **74.50** | **83.61** | **60.38** | **71.83** |

Table 3: The results highlight our Thin-Thick Adapter's superior performance over other methods, reaffirming the challenge of achieving satisfactory results with limited annotated thin scans using either supervised methods or fine-tuning.

We performed a third ablation study to investigate the impact of different ranges of the CPS loss hyper-parameter $\lambda$, which was fixed after various epochs $\epsilon$, aiming to assess the method's robustness. The results are presented in Table 4, indicating that adjusting $\lambda$ has a minor impact on performance. Furthermore, the hyper-parameter used in our comparative experiments, specifically $\lambda = 0 \rightarrow 0.5$ and fixed after 500 epochs, remains the optimal configuration among others.

|  | CQ500-Thin | | | | ROTEM-Thin | | | | Both | | | |
|---|---|---|---|---|---|---|---|---|---|---|---|---|
|  | mDSC | aDSC | mIoU | aIoU | mDSC | aDSC | mIoU | aIoU | mDSC | aDSC | mIoU | aIoU |
| $\epsilon = 0, \lambda = 0.5$ | 76.93 | 83.01 | 63.98 | 71.26 | 58.83 | 75.12 | 46.91 | 60.03 | 75.70 | 80.31 | 62.83 | 67.58 |
| $\epsilon = 300, \lambda = 0 \rightarrow 0.5$ | 77.48 | 83.49 | 64.09 | 71.73 | 59.31 | 75.48 | 47.53 | 60.67 | 76.62 | 80.94 | 63.21 | 68.34 |
| $\epsilon = 700, \lambda = 0 \rightarrow 0.5$ | 77.51 | 83.37 | 64.12 | 71.79 | 59.33 | 75.36 | 47.48 | 60.55 | 76.58 | 80.91 | 63.17 | 68.29 |
| $\epsilon = 500, \lambda = 0 \rightarrow 0.3$ | 76.50 | 81.76 | 62.99 | 69.37 | 58.45 | 74.84 | 46.32 | 59.78 | 74.81 | 80.12 | 61.13 | 66.74 |
| $\epsilon = 500, \lambda = 0 \rightarrow 0.7$ | 77.13 | 82.66 | 63.41 | 70.08 | 59.51 | 75.46 | 47.01 | 60.38 | 75.95 | 81.11 | 63.39 | 67.84 |
| $\epsilon = 500, \lambda = 0 \rightarrow 0.5$ | **77.54** | **83.62** | **64.33** | **71.86** | **59.69** | **75.64** | **47.68** | **60.82** | **76.94** | **81.36** | **63.45** | **68.57** |

Table 4: The ablations of different ranges of hyperparameters of the CPS loss, $\lambda$ and $\epsilon$, indicates the robustness of the proposed method. The optimal configuration is observed with hyperparameters $\lambda = 0 \rightarrow 0.5$ and $\epsilon = 500$.

## 7 CONCLUSION

In conclusion, medical imaging segmentation faces a challenge when working with thinner CT slices due to the limited availability of annotated data for supervision. This study addresses this issue by introducing three significant contributions. Firstly, we propose segmenting thin slices using annotations originally designed for thicker slices, thus enhancing the clinical utility of thin slice data. Secondly, we introduce the CQ500-Thin dataset, which includes thin CT scans with expert annotations, providing a valuable benchmark for future research. Finally, we present the Thin-Thick Adapter, a robust pipeline that aligns thick and thin slices and employs 3D-CPS for unsupervised domain adaptation, achieving superior results in thin slice segmentation. These contributions not only advance the field of medical imaging segmentation but also reduce the annotation time required for thin scans.

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
