# OpenReview forum: "Thin-Thick Adapter: Segmenting Thin Scans Using Thick Annotations"
_ICLR.cc/2024/Conference — Submitted to ICLR 2024_

### Official Review · Reviewer_oEQ1 · 2023-10-27

**Soundness:** 2 fair
**Presentation:** 3 good
**Contribution:** 2 fair
**Rating:** 5
**Confidence:** 4

**Summary:**

This paper presents a strategy to segment high resolution CT images (< 1mm in the Z-axis) using annotated images from low resolutions scans. To that end, the work makes use of a cross pseudo supervision loss to train a neural network using only low resolution data. The whole strategy is denoted thick thin adapted (TTA). The paper also introduces a "thick slice database".

**Strengths:**

- the paper is clear
- Good figures to illustrate the method

**Weaknesses:**

The main criticism to this work is that it introduces a terminology that does not exist in the medical imaging community (thick and thin slices) and provides some context around it that is highly inaccurate. There is not such thing as generating thick slices from thin slices. Voxel spacing and in particular the resolution within the z axis of a scan strongly depends on the image acquisition and reconstruction process, including the properties of the scan used. In reality, both in clinical practice and research it is much more desirable to have high resolution (what here is denoted as thin slices). This is not often possible due to constraints in the acquisition process, but always desired. Hence, there is no such thing as generating "thick slices" from "thin" ones.

On the methods side, the contributions are marginal. The proposed approach consists on resampling thick images to make them thin. Then a network is trained by using a loss (cross pseudo supervision) that has previously been proposed in the literature. This work extends it to 3D, which seems straightforward, and use of it to train the model.

**Questions:**

1) As mentioned in the weakness section, this work addresses a problem and describes a scenario that does not really exist within the medical imaging field. Perhaps, the described setup may be relevant in other domains. A quick exploration of the literature may help to identify use cases where the described limitations exist and, thus, the proposed solution is relevant.
2) Figure 3 refers to some acronyms that are not introduced in the text.

**Details Of Ethics Concerns:**

The paper involves use of medical images acquired from human subjects. There should be a mention of the agreement to participate from the subjects.

---

> ### Author Response · Authors · 2023-11-19
> **Response to Reviewer oEQ1 (1/3)**
>
> **Q1.1. The main criticism to this work is that it introduces a terminology that does not exist in the medical imaging community (thick and thin slices) and provides some context around it that is highly inaccurate.**
>
> **A1.1.:** Thanks for the review! The terminology of thin slices and thick slices [1,2,3,15,17], also refers to thin sections and thick sections [4,5,6,16,18], is a widely used terminology in the radiology community and in medical imaging analysis. We believe there is a large body in the clinical and computer vision community that do utilise the terminology of thick and thin slice CT imaging. For this community it is an appreciated problem that pixel level annotations of thin slice CT data is extremely time consuming. The TTA provides a practical means of addressing this issue. We have adapted a model trained on thick slice data to accurately segment on thin slice data. These thin slice segmentations can then be used in various research based applications. Reliance on manual segmentations to produce such annotations, delays research advancements and we believe provides strong impetus for reviewers to consider the TTA as a novel and citable research tool.
>
> **Q1.2. There is not such thing as generating thick slices from thin slices. Voxel spacing and in particular the resolution within the z axis of a scan strongly depends on the image acquisition and reconstruction process, including the properties of the scan used.**
>
> **A1.2.:** Many thanks for bring out the question about generate thick slices from thin slices. There are several methods to obtain thick slices as mentioned in Section 1:
>
> - One is directly from the primary reconstruction of projection data (also refers to sinogram or raw data [7]) during image acquisition [4].
>
> - The other method involved during post-processing, axial thick slices can be resliced from axial thin slices which were obtained from primary reconstruction, without having access to the projection data [8,9].
>
>   - Here is a concrete example [8]: “The DICOM files were imported into the software package Mimics 18.0 (Materialise). The initial file created from the 0.625 mm scan was then resliced to 1 mm, 1.25 mm, 2 mm, 2.5 mm and 5 mm for each head. The FOV and pixel size remained consistent through this process.”
>   - Here is another example [9]: “The range of the distal femur through the proximal tibia was covered. Images were obtained with use of 0.625 mm of slice increment, 110 kV 230 mAs, and in-plane resolution of 512 × 512. … The original CT and MRI data were imported into Mimics (version 14.1, Materialise, Leuven, Belgium). The imported images were first reconstructed into volume and then resliced to create a new axial image set with the slice increment of 1.5 mm.”
>
> - Axial thin slices allow for obtaining not only axial thick slices but also non-axial (sagittal, coronal) thick slices through multiplanar reformation (MPR, also known as multiplanar reconstruction) [4]. There are various algorithms used to thicken MPRs, including Average Intensity Projection (AIP), Maximum Intensity Projection (MIP), Minimum Intensity Projection (MinIP), and others [4]. These algorithms are applied in different scenarios, for instance, MIP is frequently utilized for visualizing bones and vascular structures [10].
>
> - Although MPR is commonly utilized to generate non-axial images from axial thin data [4], axial thick data can also be obtained through axial multiplanar reformation [11,12,13,19,20,21,22].
>
>   - Here is a concrete example [11]: “Twenty-six images of a 7.5-mm slice thickness were reconstructed from the data sets of 2.5-mm-thick and 1.25-mm-thick interval images (159 images for each phase) by a radiology technologist on a workstation (Advantage Windows 3.1; General Electric Medical Systems, Milwaukee, WI) using the axial multiplanar reformation technique.”
>
>   - Here is another example [14]: ”The resulting thin slices were then postprocessed to obtain 1-cm thick slices by using three different algorithms. The first algorithm, maximum intensity projection (MIP), generates a thick slice from a set of thin slices in which the resulting pixels represent the brightest voxels of each stack of slices.”
>
> Furthermore, there would be a notable advantage in exclusively using annotations on axial thick slices regardless they are generated either from primary reconstruction or thin slices. The TTA we proposed would make establishing ground truth annotations less resource-intensive, given that accurate segmentation of thin slice scans could be achieved using the annotations from thick slices.

---

> ### Author Response · Authors · 2023-11-19
> **Response to Reviewer oEQ1 (2/3)**
>
> **Q2. Figure 3 refers to some acronyms that are not introduced in the text.**
>
> **A2.:** My appreciation for your review, and the we will describe acronyms mentioned in Figure 3 as the following three points
>
> - In Figure 3, the left side displays various orientation views of CT scans, including axial, sagittal, and coronal, as mentioned in Section 1 Introduction, with reference.
>
> - At the bottom, there are legends depicting different semantics corresponding to various categories in brain hemorrhage, including Epidural Hemorrhage (EDH), Intraparenchymal Hemorrhage (IPH), Intraventricular Hemorrhage (IVH), Subarachnoid Hemorrhage (SAH), and Subdural Hemorrhage (SDH), as mentioned in Section 4 Dataset.
>
> - At the top, there are images, ground truth (GT), and various established models, as discussed in Section 6.1 Experiment Setup.
>
> **References**
>
> [1] Bennink, E., Oosterbroek, J., Horsch, A.D., Dankbaar, J.W., Velthuis, B.K., Viergever, M.A. and de Jong, H.W.A.M. (2015). Influence of Thin Slice Reconstruction on CT Brain Perfusion Analysis. PLOS ONE, 10(9), p.e0137766. doi:https://doi.org/10.1371/journal.pone.0137766.
>
> [2] Liu, X., Faes, L., Kale, A.U., Wagner, S.K., Fu, D.J., Bruynseels, A., Mahendiran, T., Moraes, G., Shamdas, M., Kern, C., Ledsam, J.R., Schmid, M.K., Balaskas, K., Topol, E.J., Bachmann, L.M., Keane, P.A. and Denniston, A.K. (2019). A comparison of deep learning performance against health-care professionals in detecting diseases from medical imaging: a systematic review and meta-analysis. The Lancet Digital Health, [online] 1(6), pp.e271–e297. doi:https://doi.org/10.1016/s2589-7500(19)30123-2.
>
> [3] Jin, L., Yang, J., Kuang, K., Ni, B., Gao, Y., Sun, Y., Gao, P., Ma, W., Tan, M., Kang, H., Chen, J. and Li, M. (2020). Deep-learning-assisted detection and segmentation of rib fractures from CT scans: Development and validation of FracNet. EBioMedicine, 62, pp.103106–103106. doi:https://doi.org/10.1016/j.ebiom.2020.103106.
>
> [4] Dalrymple, N.C., Prasad, S.R., Freckleton, M.W. and Chintapalli, K.N. (2005). Introduction to the Language of Three-dimensional Imaging with Multidetector CT. RadioGraphics, 25(5), pp.1409–1428. doi:https://doi.org/10.1148/rg.255055044.
>
> [5] Lee, K.H., Hong, H., Hahn, S., Kim, B., Kim, K.J. and Kim, Y.H. (2007). Summation or Axial Slab Average Intensity Projection of Abdominal Thin-section CT Datasets: Can They Substitute for the Primary Reconstruction from Raw Projection Data? Journal of Digital Imaging, 21(4), pp.422–432. doi:https://doi.org/10.1007/s10278-007-9067-y.
>
> [6] Gollub, M.J. (2002). Virtual colonoscopy. The Lancet, 360(9338), p.964. doi:https://doi.org/10.1016/s0140-6736(02)11121-4.
>
> [7] Meng, B., Wang, J. and Xing, L. (2010). Sinogram preprocessing and binary reconstruction for determination of the shape and location of metal objects in computed tomography (CT). Medical Physics, 37(11), pp.5867–5875. doi:https://doi.org/10.1118/1.3505294.
>
> [8] Ford, J.M. and Decker, S.J. (2016). Computed tomography slice thickness and its effects on three-dimensional reconstruction of anatomical structures. Journal of Forensic Radiology and Imaging, 4, pp.43–46. doi:https://doi.org/10.1016/j.jofri.2015.10.004.
>
> [9] Li, Y., Chen, C., Duan, X., Deng, B., Xiong, R., Wang, F. and Yang, L. (2015). Influence of the image levels of distal femur on the measurement of tibial tubercle-trochlear groove distance—a comparative study. Journal of Orthopaedic Surgery and Research, 10(1). doi:https://doi.org/10.1186/s13018-015-0323-4.
>
> [10] Kwon, O., Kang, S.-T., Kim, S.-H., Kim, Y.-H. and Shin, Y.-G. (2015). Maximum intensity projection using bidirectional compositing with block skipping. Journal of X-Ray Science and Technology, 23(1), pp.33–44. doi:https://doi.org/10.3233/xst-140468.
>
> [11] Kawata, S., Murakami, T., Kim, T., Hori, M., Federle, M.P., Seishi Kumano, Sugihara, E., Makino, S., Nakamura, H. and Kudo, M. (2002). Multidetector CT: Diagnostic Impact of Slice Thickness on Detection of Hypervascular Hepatocellular Carcinoma. American Journal of Roentgenology, 179(1), pp.61–66. doi:https://doi.org/10.2214/ajr.179.1.1790061.
>
> [12] Kligerman, S. and Abbott, G.F. (2010). A Radiologic Review of the New TNM Classification for Lung Cancer. American Journal of Roentgenology, 194(3), pp.562–573. doi:https://doi.org/10.2214/ajr.09.3354.
>
> [13] Kligerman, S.J., Lahiji, K., Galvin, J.R., Stokum, C. and White, C.S. (2014). Missed Pulmonary Emboli on CT Angiography: Assessment With Pulmonary Embolism–Computer-Aided Detection. American Journal of Roentgenology, 202(1), pp.65–73. doi:https://doi.org/10.2214/ajr.13.11049.
>
> [14] Diekmann, F., Meyer, H., Diekmann, S., Puong, S., Muller, S., Bick, U. and Rogalla, P. (2007). Thick Slices from Tomosynthesis Data Sets: Phantom Study for the Evaluation of Different Algorithms. Journal of Digital Imaging, 22(5), pp.519–526. doi:https://doi.org/10.1007/s10278-007-9075-y.

---

> ### Author Response · Authors · 2023-11-19
> **Response to Reviewer oEQ1 (3/3)**
>
> **References (cont.)**
>
> [15] Bajpai, V., Kyung Ho Lee, Kim, B., Kil Joong Kim, Tae Jung Kim, Young Hoon Kim and Heung Sik Kang (2008). Differences in Compression Artifacts on Thin- and Thick-Section Lung CT Images. American Journal of Roentgenology, 191(2), pp.W38–W43. doi:https://doi.org/10.2214/ajr.07.3350.
>
> [16] Guchlerner, L., Wichmann, J.L., Tischendorf, P., Albrecht, M., Vogl, T.J., Wutzler, S., Ackermann, H., Eichler, K. and Frellesen, C. (2018). Comparison of thick- and thin-slice images in thoracoabdominal trauma CT: a retrospective analysis. European Journal of Trauma and Emergency Surgery. doi:https://doi.org/10.1007/s00068-018-1021-9.
>
> [17] Kamalian, S., Atkinson, W.L., Florin, L.A., Pomerantz, S.R., Lev, M.H. and Romero, J.M. (2014). Emergency department CT screening of patients with nontraumatic neurological symptoms referred to the posterior fossa: comparison of thin versus thick slice images. Emergency Radiology, 21(3), pp.251–256. doi:https://doi.org/10.1007/s10140-014-1194-4.
>
> [18] Lee, H.Y., Goo, J.M., Lee, H.J., Lee, C.H., Park, C.M., Park, E.-A. . and Im, J.-G. . (2009). Usefulness of concurrent reading using thin-section and thick-section CT images in subcentimetre solitary pulmonary nodules. Clinical Radiology, 64(2), pp.127–132. doi:https://doi.org/10.1016/j.crad.2008.09.003.
>
> [19] Eifer, M., Tau, N., Alhoubani, Y., Kanana, N., Domachevsky, L., Shams, J., Keret, N., Gorfine, M. and Eshet, Y. (2021). COVID-19 mRNA Vaccination: Age and Immune Status and Its Association with Axillary Lymph Node PET/CT Uptake. Journal of Nuclear Medicine, 63(1), pp.134–139. doi:https://doi.org/10.2967/jnumed.121.262194.
>
> [20] Hinrichs, J.B., Christian von Falck, Hoeper, M.M., Olsson, K.M., Wacker, F., Meyer, B. and Renne, J. (2016). Pulmonary Artery Imaging in Patients with Chronic Thromboembolic Pulmonary Hypertension: Comparison of Cone-Beam CT and 64-Row Multidetector CT. Journal of Vascular and Interventional Radiology, 27(3), pp.361-368.e2. doi:https://doi.org/10.1016/j.jvir.2015.11.046.
>
> [21] Karmonik, C., Müller-Eschner, M., Partovi, S., Geisbüsch, P., Ganten, M.-K., Bismuth, J., Davies, M.G., Böckler, D., Loebe, M., Lumsden, A.B. and Tengg-Kobligk, H. von (2013). Computational Fluid Dynamics Investigation of Chronic Aortic Dissection Hemodynamics Versus Normal Aorta. Vascular and Endovascular Surgery, 47(8), pp.625–631. doi:https://doi.org/10.1177/1538574413503561.
>
> [22] Tsili, A.C. and Argyropoulou, M.I. (2015). Advances of multidetector computed tomography in the characterization and staging of renal cell carcinoma. World Journal of Radiology, [online] 7(6), pp.110–127. doi:https://doi.org/10.4329/wjr.v7.i6.110.

---

> ### Comment · Reviewer_oEQ1 · 2023-11-21
> **Answer to rebuttal**
>
> Many thanks to the authors for their efforts in providing this very detailed rebuttal.
>
> The responses have clarified aspects that were not clear to me before. Indeed, there is a difference between the slice thickness that a scanner can produce and that one from the reconstruction. This work is targeting the second case. The first case, as I pointed out in my review, is beyond reach and one may want to have the potential to have slices as thin as possible (while minimising the radiation exposure). This is usually associated with a very powerful scan (its collimator). What the authors aim to address is now clear to me.
>
> I would have liked to see more comments regarding the novelty of the work. For instance, there are open-source tools, like 3D Slicer, that offer tools to interpolate slices, i.e. Thick Slab reconstruction. In this case, one obtains the best possible slices in terms of thickness, uses slicer to get thick slides and annotates there. In which way is it better to use your approach than that one offered by 3D slicer?
>
> The Unet is designed for sparse annotations, which could be analogous to your slice thickness approach. Hence, your results would be more potent if you compared them to nnUnet in one of the public benchmarks where it has achieved a very high DSC (perhaps winning the challenge).  You should use those results as a baseline and then show different results on the configurations you propose. Using a benchmark that has been recognised by the community makes your case stronger than presenting results on a dataset that, as of now, only you have access to.

---

> ### Author Response · Authors · 2023-11-21
> **Follow-up Response to Reviewer oEQ1 (1/3)**
>
> My deep appreciation for the prompt response and follow-up discussion. Your support is invaluable to our work.
>
> **Q3. I would have liked to see more comments regarding the novelty of the work. For instance, there are open-source tools, like 3D Slicer, that offer tools to interpolate slices, i.e. Thick Slab reconstruction. In this case, one obtains the best possible slices in terms of thickness, uses slicer to get thick slides and annotates there. In which way is it better to use your approach than that one offered by 3D slicer?**
>
> **A3.:** Thank you for initiating this discussion comparing Thin-Thick Adapter (TTA) to 3D slicer and interpolation techniques. Before delving into this question, it's crucial to reiterate our primary objective, as mentioned in the paper, which is to use annotations on thick slices to perform segmentation on thin slices **without requiring any thin-thick pair**. This means the **thick annotation can be sourced from any public thick annotated dataset**, and **it doesn't have to be the exact thick slices generated from your thin slices**. And the mention of thick slice acquisition in Section 1 is just simply to introduce how thick slices are obtained in clinical practice.
>
> As you pointed out, there are open-source tools like 3D Slicer [1] that enable users to generate thick slices from thin slices through **Thick Slab Reconstruction (TRS)** [2]. Additionally, 3D Slicer can interpolate thick slices to obtain upsampled thin slices using built-in interpolation algorithms in the **Resample Image** [3] of 3D Slicer. Specifically, Thick Slab Reconstruction (TRS) employs the exact method mentioned in **A1.2**, which is utilizing multiplanar reformation (MPR), such as Maximum Intensity Projection (MIP), to obtain thick slices from thin slices [2]. And the Resample Image function in 3D slicer incorporates multiple interpolation algorithms, such as linear interpolation, to upsample thick slices into interpolated thin slices [3].
>
> Let's consider the scenario you mentioned where users obtain thick slices from thin slices using Thick Slab Reconstruction, and annotate these thick slices, then interpolate the annotated thick slices to obtain upsampled annotated thin slices using interpolation algorithms such as linear interpolation in the Resample Image function. However, this approach has two significant drawbacks:
>
> - The method still requires users to pixel-level annotate the thick scans, which is not necessary in our TTA. The thick annotations used in TTA can come from any publicly annotated thick dataset, such as BHSD [4] for brain hemorrhage, HECKTOR [5] for head and neck tumors, HaN-Seg [6] for organs-at-risk, and numerous others. With the development of the medical imaging community, obtaining annotated thick data is possible from public datasets in most cases, without the need for manual annotation.
>
> - Since the upsampled thin scans are obtained by interpolating thick scans, there is a substantial loss of information in details compared to the original thin slices, resulting in inaccurate annotations. In contrast, our Thin-Thick Adapter operates directly on thin slices, avoiding information loss and enabling more accurate segmentation.
>
> You may also would like to mentioned that there are extensions in 3D Slicer [1], such as TotalSegmentator [7], MONAI Label [8], and Nvidia AI-assisted annotation (AIAA) [9], which can assist users in automatically annotating thick scans. However, it's important to acknowledge two significant drawbacks in this approach:
>
> - The extensions used for automatic segmentation in 3D Slicer typically serve as basic assistance tools for segmentation. Although some extensions, like MONAI Label [8], employ deep learning techniques, they are not specifically fine-tuned for a particular segmentation task and lack tunability. Consequently, they perform segmentation directly on scans without any adaptation, which falls short compared to a network like nnUNet [10] that can be trained for a specific task, not to mention when compared to our TTA.
>
> - This approach shares the same issue as the one in the first scenario. The upsampled annotated thin slices, obtained through interpolation from annotated thick slices, experience substantial information loss compared to the original thin slices, resulting in inaccurate segmentation.
>
> You may also consider initially interpolating annotated thick scans into upsampled annotated thin slices using 3D Slicer, then utilize these interpolated scans to train a model (e.g., nnUNet [10]) for segmentation on your thin slices. To compare with this approach, we conducted various experiments in Section 6.1, including comparisons with interpolation methods like trilinear interpolation [11] and UniRes [12]. The results presented in Table 1 demonstrate that our method significantly outperforms these interpolation methods, and the visualization is shown in Figure 3.

---

> ### Author Response · Authors · 2023-11-21
> **Follow-up Response to Reviewer oEQ1 (2/3)**
>
> **Q4. The Unet is designed for sparse annotations, which could be analogous to your slice thickness approach. Hence, your results would be more potent if you compared them to nnUnet in one of the public benchmarks where it has achieved a very high DSC (perhaps winning the challenge). You should use those results as a baseline and then show different results on the configurations you propose. Using a benchmark that has been recognised by the community makes your case stronger than presenting results on a dataset that, as of now, only you have access to.**
>
> **A4.:**  My appreciation for this suggestion. As you mentioned, U-Net [13], especially 3D U-Net [14], serves two main purposes: (1) conducting dense volumetric segmentation using sparse annotations from orthogonal xy, xz, and yz slices in each volume [14], and (2) extending the U-Net architecture to 3D convolution for performing 3D segmentation [14].
>
> However, segmenting thin slices solely with thick annotations differs significantly from segmenting volumes with sparse annotations on orthogonal xy, xz, and yz slices. Unlike the sparse annotation approach mentioned in 3D U-Net [14], axial thick scans are not merely a selection of slices from axial thin scans, and do not contain both orthogonal xy, xz, and yz slices, but only including axial reconstruction. If we have to draw a comparison with concepts in 3D U-Net, the axial plane is similar to the xy plane.
>
> Additionally, as you recommended, we need to compare our TTA with modern models utilizing a 3D U-Net backbone, like nnUNet [10]. And we conducted the experiment in Section 6.1, and the results in Table 1 demonstrate that our method has significantly outperformed nnUNet.
>
> Moreover, you recommended using nnUNet as a baseline and comparing it in established public benchmarks that have been acknowledged by the community. We fully support this suggestion as it would help us validate the robustness of our method. However, as discussed in **A4** to **Reviewer WdYo**, finding a publicly available thin-slices dataset with a pixel-level annotated evaluation set for assessing our TTA is currently challenging. The lack of widespread attention and recognition for thin-slice segmentation within the AI for health community underscores a significant gap between computer vision for medical imaging and real-world clinical practice. This gap motivated us to propose this important research topic and create a newly annotated dataset to advance knowledge and benefit the entire community.
>
> Furthermore, to enhance diversity in our evaluation, we select brain hemorrhage, which is a inherently diverse condition as discussed in **A1** to **Reviewer WdYo**. Additionally, for further diversity, our evaluation datasets encompass both spontaneous brain hemorrhage (CQ500-Thin) and traumatic brain hemorrhage (ROTEM-Thin), significantly contributing the robustness of our proposed method. Our goal is to engage with the global research community in this field, seeking external validation of the model and promoting additional research.
>
> Once again, I sincerely appreciate your valuable suggestions and support for our work!

---

> > ### Author Response · Authors · 2023-11-21
> > **Follow-up Response to Reviewer oEQ1 (3/3)**
> >
> > **References**
> >
> > [1] Fedorov, A., Beichel, R., Kalpathy-Cramer, J., Finet, J., Fillion-Robin, J.-C., Pujol, S., Bauer, C., Jennings, D., Fennessy, F., Sonka, M., Buatti, J., Aylward, S., Miller, J.V., Pieper, S. and Kikinis, R. (2012). 3D Slicer as an image computing platform for the Quantitative Imaging Network. Magnetic Resonance Imaging, 30(9), pp.1323–1341. doi:https://doi.org/10.1016/j.mri.2012.05.001.
> >
> > [2] Volumes — 3D Slicer documentation. [online] Available at: https://slicer.readthedocs.io/en/latest/developer_guide/script_repository/volumes.html.
> >
> > [3] Resample Image (BRAINS) — 3D Slicer documentation. [online] Available at: https://slicer.readthedocs.io/en/latest/user_guide/modules/brainsresample.html.
> >
> > [4] Wu, B., Xie, Y., Zhang, Z., Ge, J., Kaspar Yaxley, Bahadir, S., Wu, Q., Li, Y. and To, M. (2023). BHSD: A 3D Multi-class Brain Hemorrhage Segmentation Dataset. Lecture Notes in Computer Science, pp.147–156. doi:https://doi.org/10.1007/978-3-031-45673-2_15.
> >
> >
> > [5] Oreiller, V., Andrearczyk, V., Jreige, M., Boughdad, S., Elhalawani, H., Castelli, J., Vallières, M., Zhu, S., Xie, J., Peng, Y., Iantsen, A., Hatt, M., Yuan, Y., Ma, J., Yang, X., Rao, C., Pai, S., Ghimire, K., Feng, X. and Naser, M.A. (2022). Head and neck tumor segmentation in PET/CT: The HECKTOR challenge. Medical Image Analysis, [online] 77, p.102336. doi:https://doi.org/10.1016/j.media.2021.102336.
> >
> > [6] Podobnik, G., Primož Strojan, Peterlin, P., Ibragimov, B. and Tomaž Vrtovec (2023). HaN‐Seg: The head and neck organ‐at‐risk CT and MR segmentation dataset. Medical Physics, 50(3), pp.1917–1927. doi:https://doi.org/10.1002/mp.16197.
> >
> > [7] Jakob Wasserthal, Breit, H.-C., Meyer, M.T., Pradella, M., Hinck, D., Sauter, A., Heye, T., Boll, D.T., Cyriac, J., Yang, S., Bach, M. and Segeroth, M. (2023). TotalSegmentator: Robust Segmentation of 104 Anatomic Structures in CT Images. Radiology. doi:https://doi.org/10.1148/ryai.230024.
> >
> > [8] Diaz-Pinto, A., Alle, S., Nath, V., Tang, Y., Ihsani, A., Asad, M., Pérez-García, F., Mehta, P., Li, W., Flores, M. and Roth, H.R., 2022. Monai label: A framework for ai-assisted interactive labeling of 3d medical images. arXiv preprint arXiv:2203.12362. doi:https://doi.org/10.48550/arxiv.2203.12362.
> >
> > [9] NVIDIA (2023). NVIDIA AI-Assisted Annotation Client. [online] Available at: https://github.com/NVIDIA/ai-assisted-annotation-client
> >
> > [10] Isensee, F., Jaeger, P.F., Kohl, S.A.A., Petersen, J. and Maier-Hein, K.H. (2020). nnU-Net: a self-configuring method for deep learning-based biomedical image segmentation. Nature Methods, 18(2), pp.203–211. doi:https://doi.org/10.1038/s41592-020-01008-z.
> >
> > [11] Thanh, C.Q. and Hai, N.T., 2017. Trilinear interpolation algorithm for reconstruction of 3D MRI brain image. American Journal of Signal Processing, 7(1), pp.1-11.
> >
> > [12] Mikael Brudfors, Yaël Balbastre, Parashkev Nachev and Ashburner, J. (2018). MRI Super-Resolution Using Multi-channel Total Variation. Communications in computer and information science, pp.217–228. doi:https://doi.org/10.1007/978-3-319-95921-4_21.
> >
> > [13] Ronneberger, O., Fischer, P. and Brox, T. (2015). U-Net: Convolutional Networks for Biomedical Image Segmentation. Lecture Notes in Computer Science, 9351, pp.234–241.
> >
> > [14] Çiçek, Ö., Abdulkadir, A., Lienkamp, S.S., Brox, T. and Ronneberger, O. (2016). 3D U-Net: Learning Dense Volumetric Segmentation from Sparse Annotation. Medical Image Computing and Computer-Assisted Intervention – MICCAI 2016, pp.424–432. doi:https://doi.org/10.1007/978-3-319-46723-8_49.

---

> ### Author Response · Authors · 2023-11-23
> **(Reminder) Kindly asking Reviewer oEQ1 for following up the rebuttal**
>
> Dear Reviewer oEQ1,
>
> My appreciation for your review, and kindly asking if you can follow up our rebuttal. We genuinely hope these follow-up rebuttal have effectively addressed your concerns. Since there are only a couple hours left, let us know your thoughts.
>
> Best regards,
>
> Authors of Submission 9133

---

### Official Review · Reviewer_KXr5 · 2023-10-31

**Soundness:** 3 good
**Presentation:** 3 good
**Contribution:** 3 good
**Rating:** 6
**Confidence:** 4

**Summary:**

This paper proposes the task of segmenting thin slices/scans directly using annotations from thicker slices (with unpaired data), as opposed to the more direct approaches of inferring thin slice annotations from thick slices as ground truth, or some combination of training/finetuning on thick slices and possibly-paired but limited thin slice annotations. The fundamental issue is a lack of annotations on thin slices, due to the costs of procuring such annotations. Instead, the proposed Thick-Thin Adapter (TTA) applies an unsupervised domain adaptation approach – that performs data alignment/augmentation (DA) with adjustable depth spacing on thick slices – before applying 3D Cross Pseudo Supervision (3D-CPS) with unlabelled thin slice data.

**Strengths:**

-	Appropriate and direct adaptation of thick-thin slice task to the 3D-CPS methodology
-	Ablation experiments to justify both DA and 3D-CPS

**Weaknesses:**

-	Relative lack of technical novelty, from direct application of existing 3D-CPS

**Questions:**

1. Annotation robustness appears especially relevant with fine-grained (and limited) data such as thin slices. Might any indication of inter-grader reliability be known, for the labelled thin slices?
2. In Section 5.1, the generation of thin slices from thick slices via duplicating and depth spacing adjustment (i.e. thinning) does not appear to take neighbouring slices into account, which seems natural at the thin slice boundaries. Might some interpolation/additional processing have been considered for the output thin slice at the original thick slice boundaries?
3. In Section 5.2, the methodology involving gradual linear increase of λ as the weight of the cross-supervision loss, is not justified in detail.

---

> ### Author Response · Authors · 2023-11-20
> **Response to Reviewer KXr5 (1/2)**
>
> **Q1. Annotation robustness appears especially relevant with fine-grained (and limited) data such as thin slices. Might any indication of inter-grader reliability be known, for the labeled thin slices?**
>
> **A1.:** My deep appreciation for your helpful suggestion of inter-grader reliability/inter-rater reliability. Combined efforts between three annotators ADH, AB and CB were used to produce the annotations.
>
> Author and board certified neuroradiologist CW initially conducted three 150 minute training sessions to teach author ADH intracranial hemorrhage identification, segmentation and classification.
>
> ADH then produced segmentations independently for 25 thin slice CT image stacks. These segmentations were then reviewed by author AB and finally CW. Following review of the segmentations feedback was provided and necessary modifications made to the segmentations. This process occurs in several iterative rounds until all authors had reached consensus that robust annotations capturing intracranial hemorrhage were produced.
>
> We believe this iterative approach involving multiple authors creating the annotations addresses the concerns related to the robustness of the thin slice annotations used as our ground truth measure.
>
> While we thank the reviewer for the suggestion of using a measure of inter-grader reliability, since the annotations were not produced independently we believe this is not appropriate in our circumstance. We believe a more robust marker of the reliability of the segmentations will be offered when the model is externally validated
>
> **Q2. In Section 5.1, the generation of thin slices from thick slices via duplicating and depth spacing adjustment (i.e. thinning) does not appear to take neighbouring slices into account, which seems natural at the thin slice boundaries. Might some interpolation/additional processing have been considered for the output thin slice at the original thick slice boundaries?**
>
> **A2.:** Thanks for the advice for taking neighbouring slices into account in data alignment. In Section 6, we conducted a comparison of interpolation methods widely used in medical imaging, such as trilinear interpolation and UniRes. The interpolated images are shown in Figure 2, and, as you noted, interpolation considers neighboring slices, resulting in a more natural appearance on sagittal and coronal views. However, two major issues limit interpolation performance. First, the interpolated mask lacks accuracy and precision, hindering overall performance. Second, interpolated images are notably blurrier than the original thick slices, diminishing the contrast of small structures and brain tissue details, consequently reducing performance.
>
> It's important to highlight that our goal is not to generate pseudo-thin slices from thick slices. Instead, we utilize thick annotations to enhance the segmentation performance of thin slices. The alignment of data with thick slices only serves as an intermediate step to support thin slice segmentation. The results in Tables 1 and 2 demonstrate that our TTA is currently the most effective solution for addressing the thin slice segmentation problem without any annotation on thin slices.

---

> ### Author Response · Authors · 2023-11-20
> **Response to Reviewer KXr5 (2/2)**
>
> **Q3. In Section 5.2, the methodology involving gradual linear increase of λ as the weight of the cross-supervision loss, is not justified in detail.**
>
> **A3.:** Thanks for pointing it out. We set λ to increase linearly from 0 to 0.5, and it remains fixed after 500 epochs, which has been updated in Section 6.1 in revised paper. Moreover, we have conducted an ablation of investigating different ranges of $\lambda$ and which was fixed after various epochs $\epsilon$ in Section 6.4. of the updated revised paper.
>
> |                          | CQ500-Thin |           |           |           | ROTEM-Thin |           |           |           | Both |           |           |           |
> |--------------------------|-----------|-----------|-----------|-----------|-----------|-----------|-----------|-----------|---------------------------|-----------|-----------|-----------|
> |                          | mDSC | aDSC | mIoU | aIoU | mDSC | aDSC | mIoU | aIoU | mDSC | aDSC | mIoU | aIoU |
> |--------------------------|------|------|------|------|------|------|------|------|------|------|------|------|
> | $\epsilon = 0, \lambda = 0.5$ | 76.93 | 83.01 | 63.98 | 71.26 | 58.83 | 75.12 | 46.91 | 60.03 | 75.70 | 80.31 | 62.83 | 67.58 |
> | $\epsilon = 300, \lambda = 0 \rightarrow 0.5$ | 77.48 | 83.49 | 64.09 | 71.73 | 59.31 | 75.48 | 47.53 | 60.67 | 76.62 | 80.94 | 63.21 | 68.34 |
> | $\epsilon = 700, \lambda = 0 \rightarrow 0.5$ | 77.51 | 83.37 | 64.12 | 71.79 | 59.33 | 75.36 | 47.48 | 60.55 | 76.58 | 80.91 | 63.17 | 68.29 |
> |--------------------------|------|------|------|------|------|------|------|------|------|------|------|------|
> | $\epsilon = 500, \lambda = 0 \rightarrow 0.3$ | 76.50 | 81.76 | 62.99 | 69.37 | 58.45 | 74.84 | 46.32 | 59.78 | 74.81 | 80.12 | 61.13 | 66.74 |
> | $\epsilon = 500, \lambda = 0 \rightarrow 0.7$ | 77.13 | 82.66 | 63.41 | 70.08 | 59.51 |75.46 | 47.01 | 60.38 | 75.95 | 81.11 | 63.39 | 67.84 |
> |--------------------------|------|------|------|------|------|------|------|------|------|------|------|------|
> | **$\epsilon = 500, \lambda = 0 \rightarrow 0.5$** | **77.54** | **83.62** | **64.33** | **71.86** | **59.69** | **75.64** | **47.68** | **60.82** | **76.94** | **81.36** | **63.45** | **68.57** |
>
> The table above displays the ablations of various ranges of hyper-parameters ($\lambda$ and $\epsilon$) in the CPS loss. The results indicate that the configuration $\lambda = 0 \rightarrow 0.5$ and $\epsilon = 500$, used in our experiments, represents the optimal hyper-parameters. Furthermore, our method demonstrates consistently good performance across different sets of hyper-parameters, with only minor fluctuations, suggesting the robustness of our approach.

---

> ### Comment · Reviewer_KXr5 · 2023-11-21
> **Answer to Rebuttal**
>
> We thank the authors for addressing our points raised in detail. Nevertheless, it might be strongly considered to attempt inter-grader analysis in future work.

---

### Official Review · Reviewer_WdYo · 2023-11-06

**Soundness:** 3 good
**Presentation:** 2 fair
**Contribution:** 2 fair
**Rating:** 5
**Confidence:** 3

**Summary:**

This paper discusses the importance of medical imaging segmentation, particularly the challenges faced when working with thin CT slices due to the lack of annotated data for model training. To overcome the difficulties in training segmentation models on thin slice data, the authors introduce three innovations: a new task for segmenting thin scans using thicker slice annotations, a dataset called CQ500-Thin with Non-Contrast CT scans and expert annotations for thin slices, and a Thin-Thick Adapter (TTA) module that significantly improves segmentation performance on thin slices, making the models more versatile for clinical use.

**Strengths:**

This paper introduces a new problem formulation that allows for the segmentation of thin slices using annotations designed for thick slices, which is beneficial for clinical applications and has not been explored much before. Additionally, this paper presents a new dataset, CQ500-Thin Dataset, with expertly labeled thin-slice CT scans, which can serve as a benchmark for future research in this field.
The proposed method employs a straightforward data alignment technique and unsupervised domain adaptation to enhance model performance on unlabeled thin slices, outperforming existing methods and showing significant improvements in standard evaluation metrics such as mean Dice Similarity Coefficient (mDSC) and mean Intersection over Union (mIoU).

**Weaknesses:**

While the newly introduced CQ500-Thin dataset offers 15 expertly labeled pixel-level annotations, this may not be representative of the diverse pathologies encountered in clinical practice. The reliance on annotations from thicker slices may not effectively capture the nuanced details inherent in thin slices. Furthermore, although the proposed method decreases the necessity for thin slice annotations, it could still be resource-intensive, posing a potential limitation in resource-constrained environments. The evaluation of this method has been confined to the specified datasets, namely CQ500-Thin and ROTEM-Thin, raising concerns about its generalizability across varied datasets or medical imaging modalities. An expanded comparative analysis would be beneficial to more comprehensively assess the method's effectiveness.

**Questions:**

Could you expand the CQ500-Thin dataset to include more than 15 expertly labeled pixel-level annotations to better represent the variety of pathologies found in clinical settings?
How might the method be adapted to capture the fine-grained details in thin slices, which may not be reflected in annotations from thick slices?
Could you discuss the potential for the proposed method's applicability to other datasets or medical imaging modalities beyond CQ500-Thin and ROTEM-Thin? Would it be possible to conduct a more extensive comparison to assess the effectiveness of the proposed method across different datasets and imaging scenarios?

---

> ### Author Response · Authors · 2023-11-21
> **Response to Reviewer WdYo (1/3)**
>
> **Q1. While the newly introduced CQ500-Thin dataset offers 15 expertly labeled pixel-level annotations, this may not be representative of the diverse pathologies encountered in clinical practice. Could you expand the CQ500-Thin dataset to include more than 15 expertly labeled pixel-level annotations to better represent the variety of pathologies found in clinical settings?**
>
> **A1.:** Thank you for raising this question, as it holds significant relevance for the potential application of the model in clinical practice.
>
> We agree that the relatively limited number of thin slice annotations used to evaluate the TTA (15 from CQ 500 and 10 from ROTEM, 25 in total) will not cover all nuances of the variable pathologies producing intracranial hemorrhage. However, this issue may be present in any brain hemorrhage segmentation dataset, whether thin or thick, and whether focused on foreground-background segmentation or semantic segmentation (e.g., BCIHM [1], INSTANCE [2], or BHSD [3]). Brain hemorrhage is a diverse condition with various types, distributions, and appearances, attributed to factors like trauma, hypertension, amyloid angiopathy, vascular malformation, and brain aneurysm rupture [9,10]. To address this diversity, our evaluation dataset includes different types and causes of brain hemorrhage. CQ500-Thin contains more instances of **spontaneous brain hemorrhage**, while ROTEM-Thin contains more **traumatic brain hemorrhage** instances from an ongoing TBI (Traumatic Brain Injury) research. In total, there are **25 pixel-level annotations** on thin datasets for evaluation. And **currently CQ500-Thin is ready to release** once the paper gets acceptance, and for ROTEM-Thin we have **already submitted a publishable application** and is undergoing ethical checking. Once the request has been approved, we **will publish the ROTEM-thin** as well.
>
> Furthermore, the BHSD dataset [3] we used for supervised training consists of a larger number of labeled thick slices (192) and number of semantics compared to other datasets like BCIHM (83) [1] and INSTANCE (130) [2].
>
> Therefore, while the number of evaluation instances impacts pathology variety, **the source of the data has a more significant impact**. The diverse brain hemorrhage causes and types in our evaluation benchmark support the Thin-Thick Adapter in better representing the range of pathologies encountered in clinical settings. **And we will expand more pixel-level annotations in further research**.
>
> **Q2. The reliance on annotations from thicker slices may not effectively capture the nuanced details inherent in thin slices. How might the method be adapted to capture the fine-grained details in thin slices, which may not be reflected in annotations from thick slices?**
>
> **A2.:** Thanks for your comment. As you pointed out, smaller lesions with small depth are more easily captured on thin slices but harder on thick slices [4]. It's important to note that small lesions with larger depth (also known as thicker lesions) instead can be captured in thick slices due to their relatively larger contrast from the axial view, even when the slices are thicker [5].
>
> The question will be addressed from following two perspectives: axial and non-axial views.
>
> - On the axial plane, the number of axial slices in a unit of volumetric space is increasing. For instance, in a space with a depth of 5mm, there is only one 5mm thick slice, but five 1mm thin slices (assuming a contiguous interval). Small lesions (with smaller depth) will be captured on these thin slices, and their appearance from the axial view is similar to small lesions (with larger depth) captured on thick slices. Since the small lesions (with larger depth) on thick slices are precisely annotated, our TTA can segment the small lesions (with smaller depth) by extending the knowledge from thick slices to thin slices.
>
> - On the non-axial plane (sagittal, coronal), as the spatial resolution (depth) increases [4], small lesions with smaller depth can be visualized from a sagittal or coronal view. Now, these small lesions are visible from both axial and non-axial views, allowing the TTA to smoothly segment these small lesions in the non-axial planes, shown in Figure 3, which is similar to their appearance in the axial plane. This capability sets our TTA apart from previous medical imaging segmentation models, such as nnUNet. And the significant improvement in the performance shows that our TTA successfully address the problem.
>
> Again, my appreciation for bringing out this significant question, since it is exactly one of the benefits of our thin-slice research topic and TTA compared to previous medical imaging segmentation tasks and models, and it is also one of the reasons why we proposed this novel research topic and method that hold considerable potential for practical applications in clinical settings.

---

> ### Author Response · Authors · 2023-11-21
> **Response to Reviewer WdYo (2/3)**
>
> **Q3. Furthermore, although the proposed method decreases the necessity for thin slice annotations, it could still be resource-intensive, posing a potential limitation in resource-constrained environments.**
>
> **A3.:** Thanks for this review. As discussed in Section 3, our method can perform segmentation on thin slices only using annotations on thick slices, without requiring any thin-thick pair. This means the thick annotation can be sourced from any public thick annotated dataset. Fortunately, advancements in the medical imaging community now provide various thick slice annotations, such as BHSD [3] for brain hemorrhage, HECKTOR [6] for head and neck tumors, HaN-Seg [7] for organs-at-risk, and numerous others. Hence, radiologists don’t need to annotate thick slices for most of the usual scenarios.
>
> Even in cases where there is no publicly available thick annotated dataset, and radiologists need to annotate from scratch, as discussed in **A4** for **Reviewer vpN7**, we still manage to reduce annotation time by 5 or 10 times when comparing thick to thin annotations.
>
> Therefore, our proposed method is not resource-intensive. And for further research, whether it involves publishing thick or thin slices, will contribute to our research topic.
>
> **Q4. The evaluation of this method has been confined to the specified datasets, namely CQ500-Thin and ROTEM-Thin, raising concerns about its generalizability across varied datasets or medical imaging modalities. An expanded comparative analysis would be beneficial to more comprehensively assess the method's effectiveness. Could you discuss the potential for the proposed method's applicability to other datasets or medical imaging modalities beyond CQ500-Thin and ROTEM-Thin? Would it be possible to conduct a more extensive comparison to assess the effectiveness of the proposed method across different datasets and imaging scenarios?**
>
> **A4.:** My appreciation for providing this constructive suggestion. I fully support the idea of exploring the applicability of our TTA on other datasets and modalities to assess its generalizability.
>
> Yet, as highlighted in **A1**, brain hemorrhage itself is a diverse condition with various types, distributions, and appearances, attributed to numerous causes. And we are actively working to enhance dataset diversity by including both spontaneous and traumatic brain hemorrhage datasets in the evaluation.
>
> Moreover, finding a publicly available thin-slices dataset with a pixel-level annotated evaluation set for assessing our TTA is currently challenging. The lack of widespread attention and recognition for thin-slice segmentation within the AI for health community underscores a significant gap between computer vision for medical imaging and real-world clinical practice. This gap motivated us to propose this important research topic and create a newly annotated dataset to advance knowledge and benefit the entire community.
>
> Despite the difficulty in finding a public thin dataset with annotations for evaluation, we are confident that TTA will exhibit optimal performance on other datasets (involving different organs or lesions) and modalities (such as MRI). Since the thin-thick problem is not specific to particular organs or lesions but is inherent in the entire field of computed tomography. Additionally, modalities like MRI undergo similar thickness adjustments as CT [8].
>
> In essence, our objective is to introduce a valuable and essential research topic, along with a simple-yet-effective method to address the problem, and also provide a dataset designed to inspire further research in thin-slice segmentation. Furthermore, we are currently collaborating with multiple institutions to annotate datasets for various lesions and organs on thin slices. Finally, with the introduction of TTA, we aim to engage with the global community of researchers interested in this field, seeking external validation of the model and encouraging further research.

---

> ### Author Response · Authors · 2023-11-21
> **Response to Reviewer WdYo (3/3)**
>
> **References**
>
> [1] Hssayeni, M., Croock, M., Salman, A., Al-khafaji, H., Yahya, Z., Ghoraani, B.: Computed tomography images for intracranial hemorrhage detection and segmentation. Intracranial hemorrhage segmentation using a deep convolutional model. Data 5(1), 14 (2020)
>
> [2] Li, X., et al.: The state-of-the-art 3D anisotropic intracranial hemorrhage segmentation on non-contrast head CT: the instance challenge. arXiv preprint arXiv:2301.03281 (2023)
>
> [3] Wu, B., Xie, Y., Zhang, Z., Ge, J., Kaspar Yaxley, Bahadir, S., Wu, Q., Li, Y. and To, M. (2023). BHSD: A 3D Multi-class Brain Hemorrhage Segmentation Dataset. Lecture Notes in Computer Science, pp.147–156. doi:https://doi.org/10.1007/978-3-031-45673-2_15.
>
> [4] Jung, H. (2021). Basic Physical Principles and Clinical Applications of Computed Tomography. Progress in Medical Physics, [online] 32(1), pp.1–17. doi:https://doi.org/10.14316/pmp.2021.32.1.1.
>
> [5] Gutschow, S.E., Walker, C.M., Martínez-Jiménez, S., Rosado-de-Christenson, M.L., Stowell, J. and Kunin, J.R. (2016). Emerging Concepts in Intramural Hematoma Imaging. RadioGraphics, 36(3), pp.660–674. doi:https://doi.org/10.1148/rg.2016150094.
>
> [6] Oreiller, V., Andrearczyk, V., Jreige, M., Boughdad, S., Elhalawani, H., Castelli, J., Vallières, M., Zhu, S., Xie, J., Peng, Y., Iantsen, A., Hatt, M., Yuan, Y., Ma, J., Yang, X., Rao, C., Pai, S., Ghimire, K., Feng, X. and Naser, M.A. (2022). Head and neck tumor segmentation in PET/CT: The HECKTOR challenge. Medical Image Analysis, [online] 77, p.102336. doi:https://doi.org/10.1016/j.media.2021.102336.
>
> [7] Podobnik, G., Primož Strojan, Peterlin, P., Ibragimov, B. and Tomaž Vrtovec (2023). HaN‐Seg: The head and neck organ‐at‐risk CT and MR segmentation dataset. Medical Physics, 50(3), pp.1917–1927. doi:https://doi.org/10.1002/mp.16197.
>
> [8] Leach, J.R., Zhu, C., Mitsouras, D., Saloner, D. and Hope, M.D. (2021). Abdominal aortic aneurysm measurement at CT/MRI: potential clinical ramifications of non-standardized measurement technique and importance of multiplanar reformation. Quantitative Imaging in Medicine and Surgery, 11(2), pp.823–830. doi:https://doi.org/10.21037/qims-20-888.
>
> [9] Cordonnier, C., Demchuk, A., Ziai, W. and Anderson, C.S. (2018). Intracerebral haemorrhage: current approaches to acute management. The Lancet, [online] 392(10154), pp.1257–1268. doi:https://doi.org/10.1016/s0140-6736(18)31878-6.
>
> [10] Tenny, S. and Thorell, W. (2020). Intracranial Hemorrhage. [online] PubMed. Available at: https://www.ncbi.nlm.nih.gov/books/NBK470242/.

---

> ### Author Response · Authors · 2023-11-23
> **(Reminder) Kindly asking Reviewer WdYo for following up the rebuttal**
>
> Dear Reviewer WdYo,
>
> My appreciation for your review, and kindly asking if you can follow up our rebuttal. We genuinely hope these rebuttal have effectively addressed your concerns. Since there are only a couple hours left, let us know your thoughts.
>
> Best regards,
>
> Authors of Submission 9133

---

### Official Review · Reviewer_vpN7 · 2023-11-06

**Soundness:** 3 good
**Presentation:** 4 excellent
**Contribution:** 2 fair
**Rating:** 6
**Confidence:** 4

**Summary:**

In this work, the authors work with CT data for segmentation. Usually along the depth dimension, the resolution is low (thick) and so most widespread annotations are for such datasets. The annotations for thin datasets (high resolution in the depth dimension) is difficult to obtain as it is burdensome. The authors propose a method which uses thick annotations and unlabeled thin scans in an unpaired and semi-supervised manner to generate high quality segmentations for thin datasets. Additionally, they also release a dataset CQ500-Thin to evaluate models on thin datasets in order to promote research in this direction.

**Strengths:**

1) The authors release a new dataset called CQ500-Thin. They have annotated 15 thin volumes from the original CQ500 thick dataset. This new dataset is useful for the community in terms of fair validation/comparison for methods developed in the future.
2) The authors propose a method to use thick annotations to generate segmentations for thin volumes. This has great potential to reduce the burden of GT-labeling for clinicians, as they can now focus on labeling thicker volumes and still get good performance on thin volumes.
3) The authors conduct experiments on two datasets, and consider a range of appropriate settings. They demonstrate the superiority of their method over several settings such as 2D segmentation (SegViT), fine-tuning nnUNet, nnUNet trained on thick, as well as nnUNet trained on thin. In each setting, the proposed TTA method achieves superior performance on both DICE and IoU.

**Weaknesses:**

1) Could the authors provide a discussion on using CPS over EMA for the semi-supervised component in their TTA method?
2) The paper seems more appropriate in the ‘datasets and benchmarks’ track.
3) The authors could consider showing an ablation study for different ranges of $\lambda$ to show the robustness of the method.
4) While the proposed method achieves good performance, could the authors discuss what is the norm in CT acquisition --- do most acquisitions generate thick datasets, or, are thin datasets more common? The significance of the authors' contribution depends on which of the two - thick or thin - is more prevalent in practice.

**Questions:**

1) Please also see weaknesses above.
2) In Eqn (2) and (4), $l_{sup}$ and $l_{cps}$ correspond to which loss? Dice and Cross-Entropy or something else?
3) In Eqn (3), $L^l_{cps}$ is not defined in later equations.
4) The authors could consider doing a study to show how their method can reduce annotation burden significantly. For example, they could consider the existing thick datasets as “thin” , and generate a corresponding “thick” dataset by reducing the resolution by 2 (so an available volume 512 x 512 x 300 would have a thick counterpart of 512 x 512 x 150). This new “thick” dataset (image + GT) could be obtained by dropping intermediate frames, or, by asking clinicians to annotate on these really coarse volumes. Results in this setting would demonstrate how it can ease the burden on annotators.

---

> ### Author Response · Authors · 2023-11-19
> **Response to Reviewer vpN7 (1/3)**
>
> **Q1.1. Could the authors provide a discussion on using CPS over EMA for the semi-supervised component in their TTA method?**
>
> **A1.1.:** Thanks for the suggestion. As mentioned in Section 6.1, we selected CPS [1,13] as our unsupervised domain adaptation (UDA) technique due to its superior performance compared to other semi-supervised techniques [2], including Entropy Minimization [3], Mean Teacher [4], and Interpolation Consistency [5] in a semi-supervised setting on the BHSD dataset. The BHSD paper [2] extensively investigated various semi-supervised methods in Section 3.4, employing 96 labeled data and 500 unlabeled data from the BHSD dataset. The results demonstrated that CPS outperformed other methods, such as Mean Teacher (which typically uses exponential moving average from the student model to the teacher model), by nearly **5%** in DSC. This significant difference in performance led us to explore the potential of CPS in our proposed TTA method.
>
> **Q1.2. The paper seems more appropriate in the ‘datasets and benchmarks’ track.**
>
> **A1.2.:** My appreciation for  suggestion. Although datasets are introduced and benchmarks are established, the primary contribution of the paper is outlining a novel approach to accurately segmenting thin slices without the time and resource expenses of manually annotating pixel by pixel, as compared to thick slice annotation. This approach aims to enhance the efficiency and feasibility of large-scale thin-slice CT segmentation. The introduced dataset and benchmark support our Thin-Thick Adapter and contribute to further research on thin-slice segmentation.
>
> **Q1.3. The authors could consider showing an ablation study for different ranges of $\lambda$ to show the robustness of the method.**
>
> **A1.3.:** Thanks for the suggestions, we conducted the third ablation which investigated the different ranges of $\lambda$ and which was fixed after various epochs $\epsilon$ in Section 6.4. of the updated revised paper.
>
> |                          | CQ500-Thin |           |           |           | ROTEM-Thin |           |           |           | Both |           |           |           |
> |--------------------------|-----------|-----------|-----------|-----------|-----------|-----------|-----------|-----------|---------------------------|-----------|-----------|-----------|
> |                          | mDSC | aDSC | mIoU | aIoU | mDSC | aDSC | mIoU | aIoU | mDSC | aDSC | mIoU | aIoU |
> |--------------------------|------|------|------|------|------|------|------|------|------|------|------|------|
> | $\epsilon = 0, \lambda = 0.5$ | 76.93 | 83.01 | 63.98 | 71.26 | 58.83 | 75.12 | 46.91 | 60.03 | 75.70 | 80.31 | 62.83 | 67.58 |
> | $\epsilon = 300, \lambda = 0 \rightarrow 0.5$ | 77.48 | 83.49 | 64.09 | 71.73 | 59.31 | 75.48 | 47.53 | 60.67 | 76.62 | 80.94 | 63.21 | 68.34 |
> | $\epsilon = 700, \lambda = 0 \rightarrow 0.5$ | 77.51 | 83.37 | 64.12 | 71.79 | 59.33 | 75.36 | 47.48 | 60.55 | 76.58 | 80.91 | 63.17 | 68.29 |
> |--------------------------|------|------|------|------|------|------|------|------|------|------|------|------|
> | $\epsilon = 500, \lambda = 0 \rightarrow 0.3$ | 76.50 | 81.76 | 62.99 | 69.37 | 58.45 | 74.84 | 46.32 | 59.78 | 74.81 | 80.12 | 61.13 | 66.74 |
> | $\epsilon = 500, \lambda = 0 \rightarrow 0.7$ | 77.13 | 82.66 | 63.41 | 70.08 | 59.51 |75.46 | 47.01 | 60.38 | 75.95 | 81.11 | 63.39 | 67.84 |
> |--------------------------|------|------|------|------|------|------|------|------|------|------|------|------|
> | **$\epsilon = 500, \lambda = 0 \rightarrow 0.5$** | **77.54** | **83.62** | **64.33** | **71.86** | **59.69** | **75.64** | **47.68** | **60.82** | **76.94** | **81.36** | **63.45** | **68.57** |
>
> The table above displays the ablations of various ranges of hyper-parameters ($\lambda$ and $\epsilon$) in the CPS loss. The results indicate that the configuration $\lambda = 0 \rightarrow 0.5$ and $\epsilon = 500$, used in our experiments, represents the optimal hyper-parameters. Furthermore, our method demonstrates consistently good performance across different sets of hyper-parameters, with only minor fluctuations, suggesting the robustness of our approach.

---

> ### Author Response · Authors · 2023-11-19
> **Response to Reviewer vpN7 (2/3)**
>
> **Q1.4. While the proposed method achieves good performance, could the authors discuss what is the norm in CT acquisition --- do most acquisitions generate thick datasets, or, are thin datasets more common? The significance of the authors' contribution depends on which of the two - thick or thin - is more prevalent in practice.**
>
> **A1.4:** This is a very meaningful question and thanks for bringing it out. To answer the question, we need to differentiate between **CT acquisition in clinical practice** and the **release of a dataset** for the medical imaging community.
>
> In clinical practice, axial thin slices are typically collected during primary reconstruction [6], with exceptions for specific occasions like fast scans or low-resolution CT. It is because a radiologist often needs to scroll through a stack of axial thin slices to localize lesions [7], especially for small lesions, which can only be captured on thin slices [8]. Additionally, thin slices enable the generation of thick slices, not only in the axial but also in non-axial (sagittal, coronal) thick slices through multiplanar reformation (MPR) without the needs of projection data [6]. This capability is elaborated further in the response to Reviewer oEQ1 question 1.2. Moreover, thin slices have broader applications, including accurate brain hemorrhage volume estimation, which is beyond the capabilities of thick slices. In contrast, thick slices are not frequently required during reconstruction and are primarily used for detecting low-contrast lesions due to their higher signal-to-noise ratio (SNR) [9]. Hence, in clinical practice, thin slices are more common, and since projection data are not directly viewed, storing thin slices is more prevalent.
>
> Regarding current public datasets in medical imaging and computer vision, there is a temporal aspect to consider. Older CT scans typically produce thick slices, and thin scans are more storage-intensive compared to thick scans. Consequently, public datasets such as RSNA Pneumonia Detection Challenge [10] predominantly consist of thick datasets. However, as CT and storage technology advances, more recent datasets like CQ500 [11] include thin slices.  However, obtaining large-scale pixel-level annotations for these thin scans remains impractical. This challenge motivates our paper.
>
> **Q2. In Eqn (2) and (4), $l_{sup}$ and $l_{cps}$  corresponds to which loss? Dice and Cross-Entropy or something else?**
>
> **A2.:** Thanks for pointing it out, both $l_{sup}$ and $l_{cps}$  are the combination of dice and cross-entropy losses, which are aligned with the default configured of nnU-Net's joint loss [12]. This has been updated in Section 5.2 of revised paper.
>
> **Q3. In Eqn (3), $L_{cps}^l$ is not defined in later equations.**
>
> **A3.:** Thanks for the suggestions, as we mentioned in Section 5.2, The cross-pseudo supervision loss, $L_{cps}$, comprises two parts: $L_{cps}^l$ and $L_{cps}^u$, incorporating the CPS loss on labeled and unlabeled datasets. This means that the $L_{cps}^l$ is calculated in the same fashion as $L_{cps}^u$, and the only modification is to switch unlabeled data to labeled data. In order to avoid ambiguity, we have updated the concrete loss formula in equation (4) and (5) of the updated revised paper.
>
> **Q4. The authors could consider doing a study to show how their method can reduce annotation burden significantly. For example, they could consider the existing thick datasets as “thin” , and generate a corresponding “thick” dataset by reducing the resolution by 2 (so an available volume 512 x 512 x 300 would have a thick counterpart of 512 x 512 x 150). This new “thick” dataset (image + GT) could be obtained by dropping intermediate frames, or, by asking clinicians to annotate on these really coarse volumes. Results in this setting would demonstrate how it can ease the burden on annotators.**
>
> **A4.:** Many thanks for reviewer providing the inspiring suggestion!  While this experiment could be considered to provide objective evidence of the reduction in annotation burden. We believe radiologists who have performed manual segmentation would accept that annotation of thick data is significantly easier than thin slices. Instead of the 1 in 2 reduction mentioned in the question, manual annotation is reduced by 5 or 10 times when comparing thick to thin segmentation.(thick slices are up to 5mm thickness and 30-50 slices, and thin slices are less than 1mm and 300-500 slices).
>
> Based on our own annotating process, we also speculate that
>
> - most of the annotation time is spent creating/adjusting the segmentation edge, not the inner (which can be filled)
> - thus, interpolation techniques do not help significantly for intermediate frames because the edges still need the name level of refinement
>
> Taking into account these factors, annotation time is expected to be approximately proportional to number of frames, whether they are derived from a thin dataset, or thick dataset with reduced resolution.

---

> ### Author Response · Authors · 2023-11-19
> **Response to Reviewer vpN7 (3/3)**
>
> **References**
>
> [1] Chen, X., Yuan, Y., Zeng, G. and Wang, J. (2021). Semi-Supervised Semantic Segmentation With Cross Pseudo Supervision. [online] openaccess.thecvf.com. Available at: https://openaccess.thecvf.com/content/CVPR2021/html/Chen_Semi-Supervised_Semantic_Segmentation_With_Cross_Pseudo_Supervision_CVPR_2021_paper.html [Accessed 19 Nov. 2023].
>
> [2] Wu, B., Xie, Y., Zhang, Z., Ge, J., Kaspar Yaxley, Bahadir, S., Wu, Q., Li, Y. and To, M. (2023). BHSD: A 3D Multi-class Brain Hemorrhage Segmentation Dataset. Lecture Notes in Computer Science, pp.147–156. doi:https://doi.org/10.1007/978-3-031-45673-2_15.
>
> [3] Grandvalet, Y. and Bengio, Y. (2004). Semi-supervised Learning by Entropy Minimization. [online] Neural Information Processing Systems. Available at: https://proceedings.neurips.cc/paper/2004/hash/96f2b50b5d3613adf9c27049b2a888c7-Abstract.html.
>
> [4] Tarvainen, A. and Valpola, H. (2017). Mean teachers are better role models: Weight-averaged consistency targets improve semi-supervised deep learning results. [online] Neural Information Processing Systems. Available at: https://proceedings.neurips.cc/paper/2017/hash/68053af2923e00204c3ca7c6a3150cf7-Abstract.html.
>
> [5] Verma, V., Kawaguchi, K., Lamb, A., Kannala, J., Solin, A., Bengio, Y. and Lopez-Paz, D. (2022). Interpolation consistency training for semi-supervised learning. Neural Networks, 145, pp.90–106. doi:https://doi.org/10.1016/j.neunet.2021.10.008.
>
> [6] Dalrymple, N.C., Prasad, S.R., Freckleton, M.W. and Chintapalli, K.N. (2005). Introduction to the Language of Three-dimensional Imaging with Multidetector CT. RadioGraphics, 25(5), pp.1409–1428. doi:https://doi.org/10.1148/rg.255055044.
>
> [7] Alexander, R.G., Waite, S., Macknik, S.L. and Martinez-Conde, S. (2020). What do radiologists look for? Advances and limitations of perceptual learning in radiologic search. Journal of Vision, 20(10), p.17. doi:https://doi.org/10.1167/jov.20.10.17.
>
> [8] Liu, X., Chen, L., Qi, W., Jiang, Y., Liu, Y., Zhang, M. and Hong, N. (2017). Thin-slice brain CT with iterative model reconstruction algorithm for small lacunar lesions detection. Medicine, 96(51), p.e9412. doi:https://doi.org/10.1097/md.0000000000009412.
>
> [9] Nashwan Karkhi Abdulkareem, Shereen Ismail Hajee, Fatiheea Fatihalla Hassan, Ilham Khalid Ibrahim, Hussein, E. and Noor Abubaker Abdulqader (2023). Investigating the slice thickness effect on noise and diagnostic content of single-source multi-slice computerized axial tomography. PubMed, [online] 16(6), pp.862–867. doi:https://doi.org/10.25122/jml-2022-0188.
>
> [10] Anouk Stein, MD, Carol Wu, Chris Carr, George Shih, Jamie Dulkowski, kalpathy, Leon Chen, Luciano Prevedello, Marc Kohli, MD, Mark McDonald, Peter, Phil Culliton, Safwan Halabi MD, Tian Xia. (2018). RSNA Pneumonia Detection Challenge. Kaggle. https://kaggle.com/competitions/rsna-pneumonia-detection-challenge
>
> [11] Chilamkurthy, S., Ghosh, R., Tanamala, S., Biviji, M., Campeau, N.G., Venugopal, V.K., Mahajan, V., Rao, P. and Warier, P. (2018). Development and Validation of Deep Learning Algorithms for Detection of Critical Findings in Head CT Scans. [online] arXiv.org. Available at: https://arxiv.org/abs/1803.05854
>
> [12] Isensee, F., Jaeger, P.F., Kohl, S.A.A., Petersen, J. and Maier-Hein, K.H. (2020). nnU-Net: a self-configuring method for deep learning-based biomedical image segmentation. Nature Methods, 18(2), pp.203–211. doi:https://doi.org/10.1038/s41592-020-01008-z.
>
> [13] Huang, Y., Zhang, H., Yan, Y. and Hassan, H. (2022). 3D Cross-Pseudo Supervision (3D-CPS): A Semi-supervised nnU-Net Architecture for Abdominal Organ Segmentation. Lecture Notes in Computer Science, pp.87–100. doi:https://doi.org/10.1007/978-3-031-23911-3_9.

---

> > ### Comment · Reviewer_vpN7 · 2023-11-21
> >
> > I thank the authors for answering my questions. For now, I retain the same scores and will move to update the scores after seeing responses from other reviewers.

---

### Meta-Review · Area_Chair_DMfw · 2023-12-06

**Metareview:**

This paper propose a new research problem for medical image segmentation, namely the segmentation of the thin slices often available in modern CT scanners, from annotations of the thicker slices typically available in older scanners -- and hence in much available annotated data. This is a relevant research problem, as annotation remains a significant cost for creating training data, and annotated data remains scarce in medical imaging. The authors also contribute a dataset annotated specifically for this task.

The main weaknesses of the paper related to:
- the communication of the problem's relevance -- several reviewers had a hard time understanding the real life relevance of the problem. This was partially addressed in the rebuttal, but shows a potential impact problem downstream
- Comparison with state-of-the-art models that one would expect to do well on this problem; in particular nn-U-net is recommended by multiple reviewers, and one reviewer details a way that this could be made feasible in practice.

Based on these concerns, I unfortunately cannot recommend the paper for publication at ICLR'24.

**Justification For Why Not Higher Score:**

The experimental validation and the clarity of the paper's motivation remain concerns, and these limit the paper's downstream potential impact.

**Justification For Why Not Lower Score:**

N/A

---

### Decision · Program_Chairs · 2024-01-16

Reject